# PRISM: PARETO-RESPONSIVE ITERATIVE SAMPLING WITH DPO FOR MULTI-OBJECTIVE PLANNING

## ABSTRACT

Many planning-style applications of large language models are inherently multi-objective. Beyond correctness, users care about efficiency and the avoidance of irrelevant or unsafe actions. Yet most alignment pipelines optimize a single scalar reward, which hides trade-offs and offers little control when secondary objectives have uncertain or deployment-specific weights. We present PRISM, a Pareto responsive framework that integrates Direct Preference Optimization. PRISM adds three components designed for offline, several convergence toward balanced solutions. First, it uses golden comparisons that isolate per-objective preferences. Second, it computes attention-style weights from deficiency diagnostics that combine loss and gradient information. Third, it applies Pareto guided sampling that orients preference pairs by cosine alignment with the current weight direction.This loop performs common-descent updates for a vector of objective deficiencies and stops at a certificate of first-order Pareto stationarity. It removes the need for online reinforcement learning, reward sweeps, or families of specialist models. On six benchmarks in question answering, coding, and mathematical reasoning, PRISM improves accuracy over strong baselines while simultaneously reducing latency and step count and driving off-domain actions to near zero. PRISM provides a principled and compute efficient recipe for robust multi-objective alignment of LLM-based planners.

## 1 INTRODUCTION

Large language models (LLMs) have demonstrated strong ability to generate complex multi-step solutions across question answering, code generation, and mathematical reasoning. Instruction tuning and chain-of-thought prompting encourage models to produce intermediate reasoning steps, which in turn improves task success (Ouyang et al., 2022; Wei et al., 2022; Wang et al., 2022). Despite these gains, most training pipelines still optimize a single reward signal such as correctness or a similar scalar proxy. A single objective obscures the fact that high-quality plans should also be efficient and safe, and improving one criterion can degrade others if those trade-offs are not explicitly modeled.

Many real-world planning problems are inherently multi-criteria. A code solution must be correct, execute with low latency, and avoid unnecessary tool calls or transformations. Dialogue agents are expected to be helpful, harmless, and honest. Traditional alignment methods such as reinforcement learning from human feedback compress heterogeneous preferences into one reward (Ouyang et al., 2022), which yields a one-dimensional target that cannot represent alternative compromises. Recent attempts at multi-objective alignment often assume fixed scalarization weights, require multiple training runs to cover different trade-offs, or rely on computationally intensive online reinforcement learning. In practice, priorities are typically clear for the primary objective and any hard constraints, while the relative importance of secondary goals is uncertain and application dependent. When the weighting of objectives is unknown, exploring the Pareto frontier is essential so that practitioners can select among policies where a gain in one dimension necessarily implies a loss in another.

There is an additional practical constraint that concerns model adaptability. Closed source systems such as GPT and Gemini provide strong general performance (OpenAI, 2025; Comanici et al., 2025), but weights are not accessible and fine-tuning options are limited to surface-level controls that do not modify parameters. When these systems encounter domains that were not well covered during

pretraining or exhibit hallucinations, direct adaptation is not available. Open-weight models such as T5 and LLaMA expose parameters for finetuning (Raffel et al., 2020; Dubey et al., 2024), which enables targeted improvements in new domains and the possibility of aligning planning behavior with domain-specific constraints. Recent open releases further indicate that well-tuned open models can match or surpass closed systems on a range of public benchmarks. For these reasons, this work focuses on training open models to high competence while explicitly balancing multiple planning objectives.

PRISM is a Pareto responsive iterative sampling framework that combines Direct Preference Optimization with multi-objective planning (Rafailov et al., 2023). It operationalizes three key ideas that our contributions later formalize: first, a preference fine-tuning scheme that jointly targets accuracy, efficiency, and error avoidance without relying on RL or multiple specialists; second, an attention style softmax weighting over objectives computed from loss–gradient deficiency diagnostics; third, Pareto-guided sampling that uses golden comparisons pairs markedly different on one objective yet nearly unchanged on others and orients them by cosine agreement with the current weight direction to suppress off-domain actions. This loop allows one finetuning run to move toward a balanced operating point while retaining the practicality of open-weight backbones. The remainder of the paper details the methodology and shows that PRISM improves accuracy while simultaneously enhancing efficiency and robustness across diverse benchmarks.

The main contributions are as follows:

- We present PRISM, a preference fine-tuning framework that jointly improves accuracy, efficiency, and error avoidance without resorting to RL or training multiple specialist models.
- We introduce an attention-style softmax weighting over objectives derived from loss–gradient deficiency diagnostics; to our knowledge this is the first use of such deficiency-aware weighting within DPO-based multi-objective planning for LLMs.
- We propose a Pareto guided sampling mechanism that orients preference pairs via cosine agreement with the current weight direction using golden comparisons, providing a simple and effective way to steer updates toward Pareto-consistent regions and to suppress off-domain actions in practice.

## 2 RELATED WORK

### 2.1 PLANNING WITH LLMS

Instruction tuning improves adherence to human intent and single-turn success, while explicit reasoning further boosts multi-step performance. Chain-of-Thought (CoT) prompting elicits intermediate steps and Self-Consistency aggregates diverse rationales to improve robustness (Ouyang et al., 2022; Wei et al., 2022; Wang et al., 2022). Reasoning is often interleaved with action and tool use: ReAct couples thought and environment interaction; Plan-and-Solve separates planning from execution; Tree-of-Thoughts searches over reasoning branches (Yao et al., 2023b; Wang et al., 2023; Yao et al., 2023a). Tool augmentation and program-aided reasoning offload computation and retrieval to external systems, reducing arithmetic and factual errors (Schick et al., 2023; Chen et al., 2022). Iterative self-improvement frameworks, such as Self-Refine and Reflexion, use model-generated feedback to revise drafts or trajectories (Madaan et al., 2023; Shinn et al., 2023). Beyond handcrafted pipelines, recent work automates agent workflow discovery or optimization (e.g., AFlow; ScoreFlow with score-aware DPO), but still optimizes a scalar signal per run and provides limited control over multi-criteria trade-offs (Zhang et al., 2024; Wang et al., 2025). Overall, most planning-style methods maximize a single objective (accuracy or a composite score), leaving efficiency and error-avoidance under-specified.

### 2.2 MULTI-OBJECTIVE OPTIMIZATION FOR ALIGNMENT

Reinforcement learning from human feedback(RLHF) collapses heterogeneous preferences into a single reward, making performance sensitive to reward design and weight choices (Ouyang et al., 2022; Bai et al., 2022). DPO replaces online RL with an offline preference-learning objective but is inherently single-objective (Rafailov et al., 2023). Multi-objective alignment has followed three routes. First, MORLHF sweeps weights or trains specialist policies to approximate a Pareto set,

incurring heavy compute and instability (Zhou et al., 2023). Second, model-mixing methods (Rewarded Soups) interpolate single-objective experts to cover the frontier, requiring many specialists and offering limited local control (Rame et al., 2023). Third, multi-objective preference optimization extends DPO: MODPO learns weight-conditioned solutions offline; MO-ODPO trains a single preference-conditional policy online to steer trade-offs at inference time, but adds conditioning and on-policy complexity (Zhou et al., 2023; Gupta et al., 2025). While Panacea (Wen et al., 2024) establishes an important line of work in multi-dimensional preference alignment for LLMs, its design assumptions (online preference vector injection, focus on generic LLM tasks, lack of explicit planning-constraint handling) render it less directly applicable to our strict planning setting. Constrained RL methods (CPO/RCPO) enforce hard constraints but remain RL-based and costly at LLM scale (Achiam et al., 2017; Tessler et al., 2018). Our work differs by staying fully offline and several training epoch: we introduce objective-isolating diagnostics, loss-and-gradient–based deficiency signals, and Pareto-guided sampling within a DPO loop to drive convergence toward a balanced Pareto-consistent policy without reward sweeping or training multiple models.

## 3 METHODOLOGY

This section provides an overview of the PRISM training framework, which integrates plan generation, objective-specific diagnostics, dynamic weight computation, and Pareto-guided preference optimization into several iterative loop. The overall pipeline, illustrated in Figure 2, begins with generating multiple candidate plans for each problem instance and assigning multi-dimensional objective scores. From these, golden comparisons are extracted to capture per-objective deficiencies, which are then transformed into dynamic weights guiding the sampling and DPO updates. By continuously recalculating these weights and aligning training with the Pareto frontier, PRISM ensures that the LLM converges toward a balanced trade-off solution that respects correctness, efficiency, and constraint satisfaction simultaneously. The objectives used in our experiments (accuracy, latency, step count, and avoidance of irrelevant actions) are representative rather than exhaustive. PRISM only requires retention of one primary objective (e.g., correctness), while any number of secondary objectives and hard constraints may be added, removed, or modified. The framework is thus scalable to arbitrary multi-objective settings as long as the objectives can be quantitatively assessed or pairwise compared. For clarity, PRISM consists of three tightly coupled components: Plan generation and golden comparisons (Section 3.1): collect multi-objective diagnostics that isolate per-objective deficiencies without updating the policy. Preference signals and weight computation (Section 3.2): fuse loss and gradient-based diagnostics into a deficiency vector and map it to a simplex of sampling weights. Adaptive sampling and Pareto-guided DPO training (Section 3.3): reweight preference pairs according to the current weight direction, perform DPO updates, and stop at a first-order Pareto-stationary compromise.

### 3.1 PLAN GENERATION AND GOLDEN COMPARISONS

Given a task instance $x$, the first step is to explore a broad space of potential solutions, because no single plan is likely to satisfy all objectives simultaneously. A generator LLM produces a finite set of candidate plans

$$Y(x) = \{y^{(1)}, y^{(2)}, \ldots, y^{(K)}\},$$

where each plan $y^{(k)}$ is a sequence of reasoning steps. A separate executor LLM executes each plan and a reward model evaluates the outcome on $n$ objectives, producing a vector

$$\mathbf{O}(x, y^{(k)}) = \left[ O_1(x, y^{(k)}), O_2(x, y^{(k)}), \ldots, O_n(x, y^{(k)}) \right]^\top.$$

Here $O_1$ measures the primary objective (e.g., correctness), $O_n$ measures a hard constraint (e.g., the negative of the wrong-step count so that larger is better), and the intermediate components measure auxiliary efficiency metrics such as the negative of time or step count.

To assess how well the current model performs on each objective, we examine differences between plans. For two plans $y, y' \in Y(x)$, their objective difference is

$$\Delta \mathbf{O}(x; y, y') = \mathbf{O}(x, y) - \mathbf{O}(x, y').$$

The sign and magnitude of each component tell us which plan is better on each objective. However, these raw differences intertwine all objectives, so we define golden comparisons to isolate individual

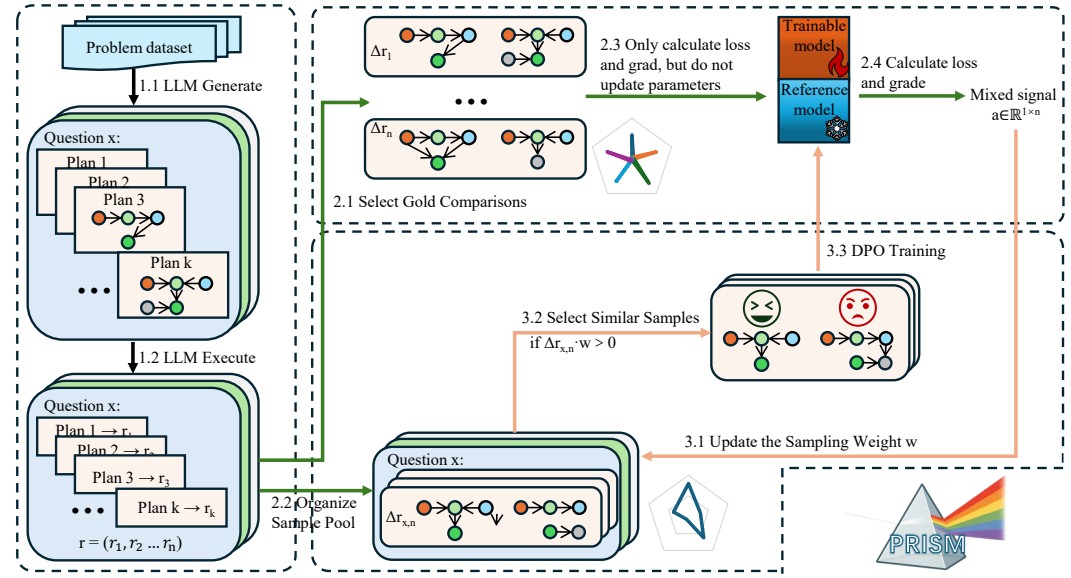

Figure 1: The pipeline of PRISM

criteria. A pair $(y, y')$ is golden for objective $i$ if

$$O_i(x, y) - O_i(x, y') > \Delta^i_{\min}, \quad \text{and} \quad |O_j(x, y) - O_j(x, y')| < \delta_j \quad \text{for all } j \neq i,$$

where $\Delta^i_{\min} > 0$ enforces that $y$ is strictly better than $y'$ on objective $i$, and $\delta_j \geq 0$ require near equality on the other objectives. These pairs serve purely as diagnostics: they reveal how well the model ranks plans with respect to a single objective while controlling for others. If there are $n$ objectives and we select $M$ pairs per objective, we obtain $n \times M$ golden comparisons. Importantly, these comparisons are not used to update the model; instead, they provide clean feedback on per-objective behaviour.

## 3.2 Preference Signals and Weight Computation

To steer the policy toward a balanced solution, we extract per-objective diagnostic signals and convert them into attention weights over objectives. Standard DPO learns from preference pairs under a single scalar objective; here we obtain a separate signal for each objective and aggregate them adaptively. After an initial warm-up phase (e.g., 100 random comparisons) to stabilize optimization, we evaluate the current policy on the golden pairs. For each golden pair $(x_i^r, y_{i,r}^+, y_{i,r}^-)$ of objective $i$ (with $r = 1, \ldots, M$), the DPO loss is

$$\mathcal{L}_{\text{DPO}}(x_i^r, y_{i,r}^+, y_{i,r}^-; \theta) = -\log \sigma\Big( \big[ s_\theta(x_i^r, y_{i,r}^+) - s_{\theta_0}(x_i^r, y_{i,r}^+) \big] - \big[ s_\theta(x_i^r, y_{i,r}^-) - s_{\theta_0}(x_i^r, y_{i,r}^-) \big] \Big), \tag{1}$$

where $s_\theta(x, y) = \log \pi_\theta(y \mid x)$ and $\sigma(z) = 1/(1 + e^{-z})$. We also compute the gradient norm

$$g_{i,r} = \big\| \nabla_\theta \mathcal{L}_{\text{DPO}}(x_i^r, y_{i,r}^+, y_{i,r}^-; \theta) \big\|_2,$$

which reflects the local difficulty of correcting the error. Averaging over golden pairs yields an objective-wise deficiency score

$$\bar{\ell}_i = \frac{1}{M} \sum_{r=1}^M \mathcal{L}_{\text{DPO}}(x_i^r, y_{i,r}^+, y_{i,r}^-; \theta), \qquad \bar{g}_i = \frac{1}{M} \sum_{r=1}^M g_{i,r}, \tag{2}$$

and we form a convex combination

$$a_i = \gamma \bar{\ell}_i + (1 - \gamma) \bar{g}_i, \qquad \gamma \in [0, 1], \tag{3}$$

so that larger $a_i$ indicates worse performance on objective $i$ (either frequent misorderings, or hard-to-fix ones).

We then turn the vector $\mathbf{a} = (a_1, \dots, a_n)$ into attention weights $\mathbf{w} \in \Delta^{n-1}$ via a temperatured softmax with a curriculum bias. Let $c_1 = +\lambda$, $c_n = -\lambda$, and $c_j = 0$ for $2 \leq j \leq n-1$, where $\lambda \in [0, 1]$ decays to zero over training (early emphasis on the primary objective, later emphasis on the hard constraint). Define standardized logits

$$z_i = \frac{a_i - \operatorname{mean}(\mathbf{a})}{\operatorname{std}(\mathbf{a}) + \epsilon} + c_i,$$

and compute

$$w_i = \frac{\exp(z_i/\tau)}{\sum_{j=1}^{n} \exp(z_j/\tau)}, \qquad \sum_{i=1}^{n} w_i = 1, \tag{4}$$

with temperature $\tau > 0$ and a small $\epsilon > 0$ for numerical stability. This attention-style normalization produces a smooth, data-driven focus: objectives with larger deficiencies receive larger weights. Crucially, golden comparisons are diagnostic only; no gradients from $\mathbf{w}$ or from these diagnostics flow into the DPO objective. The weights $\mathbf{w}$ influence sampling and pair orientation downstream, succinctly summarizing which objectives currently merit more updates without directly biasing the training loss.

### 3.3 ADAPTIVE SAMPLING, DPO UPDATE, AND PARETO-GUIDED TRAINING

The weight vector $\mathbf{w}$ not only determines how we sample comparisons from the pool but also fixes the orientation of each pair before a DPO update. For any candidate pair $(x, y, y')$ with objective-gap vector $\Delta \mathbf{O}(x; y, y') = \mathbf{O}(x, y) - \mathbf{O}(x, y')$, we compute the cosine alignment

$$s(x; y, y') \;=\; \frac{\mathbf{w}^\top \Delta \mathbf{O}(x; y, y')}{\|\mathbf{w}\|_2 \|\Delta \mathbf{O}(x; y, y')\|_2} \;\in [-1, 1].$$

The quantity $s$ measures directional agreement between the current priority $\mathbf{w}$ and the pair's gap vector. We sample pairs in proportion to the absolute alignment so that comparisons more aligned with the current deficiency receive higher probability. Concretely, denoting the pool by $\mathcal{P}$, the sampling probability is

$$q(x; y, y') \;=\; \frac{\left|s(x; y, y')\right|^\beta}{\sum_{(\tilde{x}, \tilde{y}, \tilde{y}') \in \mathcal{P}} \left|s(\tilde{x}; \tilde{y}, \tilde{y}')\right|^\beta}, \tag{5}$$

where $\beta > 0$ controls the sharpness. Orientation is determined by the sign of $s$. When $s(x; y, y')$ closed to 1, the pair is strongly aligned and $y$ is preferred over $y'$ under $\mathbf{w}$; when $s(x; y, y')$ closed to -1, the situation reverses. Formally, letting $(y^+, y^-)$ denote the ordered pair used in DPO,

$$(y^+, y^-) \;=\; \begin{cases} (y, y'), & \text{if } s(x; y, y') \geq 0, \\ (y', y), & \text{if } s(x; y, y') < 0. \end{cases}$$

Given a minibatch $S$ sampled according to the distribution $q$ in Eq. equation 5 and oriented as above, we update $\theta$ by minimizing the average DPO loss in Eq. equation 1 over $S$. Using the deficiency scores $a_i$ from Eq. equation 3, we collect them into $\mathbf{a}(\theta) = (a_1(\theta), \dots, a_n(\theta))^\top$, and obtain the corresponding sampling weights $\mathbf{w} = W(\mathbf{a}(\theta)) \in \Delta^{n-1}$ via Eq. equation 4. One outer iteration of PRISM is then written as

$$\theta^+ = T(\theta, W(\mathbf{a}(\theta))), \qquad \mathbf{a}^+ = \mathbf{a}(\theta^+), \tag{6}$$

where $T$ denotes one (or a few) gradient steps on the DPO objective using pairs drawn according to $\mathbf{w}$. For a small learning rate, the update direction is well approximated by a convex combination of objective-wise descent directions:

$$d(\theta; \mathbf{w}) \approx -\sum_{i=1}^{n} w_i \nabla a_i(\theta), \qquad \theta^+ \approx \theta + \eta \, d(\theta; \mathbf{w}).$$

Fix the current parameter $\theta$ and consider any weight vector $\mathbf{w} \in \Delta^{n-1}$. For each coordinate $i$,

$$a_i(\theta + \eta \, d(\theta; \mathbf{w})) \approx a_i(\theta) - \eta \sum_{j=1}^{n} w_j \nabla a_i(\theta)^\top \nabla a_j(\theta).$$

Writing $g_i(\theta) = \nabla a_i(\theta)$ and $G(\theta) = [g_1(\theta), \ldots, g_n(\theta)] \in \mathbb{R}^{p \times n}$, we have

$$\nabla a_i(\theta)^\top d(\theta; \mathbf{w}) = -g_i(\theta)^\top G(\theta)\, \mathbf{w}.$$

If there exists $\mathbf{w}$ such that $g_i(\theta)^\top G(\theta)\, \mathbf{w} > 0$ for all $i$, then moving along $d(\theta; \mathbf{w})$ simultaneously reduces all coordinates of $\mathbf{a}$ to first order, which means a better sampling weight still exists. Conversely, if for some tolerance $\varepsilon > 0$ one has for all $\mathbf{w} \in \Delta^{n-1}$

$$\max_i \left\{ -\nabla a_i(\theta)^\top d(\theta; \mathbf{w}) \right\} \leq \varepsilon,$$

then no convex combination of objective gradients yields a uniformly improving direction beyond $\varepsilon$. This condition is the first-order certificate that no strictly better sampling weight exists at $\theta$.

The above criterion is equivalent to Pareto stationarity. Since $d(\theta; \mathbf{w}) = -G(\theta)\mathbf{w}$ and

$$\left\| G(\theta)\mathbf{w} \right\|^2 = \sum_{i,j} w_i w_j\, g_i(\theta)^\top g_j(\theta),$$

the existence of $\mathbf{w}^\star$ with $\|G(\theta)\mathbf{w}^\star\| \leq \kappa\varepsilon$ implies that the zero vector lies, up to $\kappa\varepsilon$, in the convex hull of $\{g_i(\theta)\}_{i=1}^n$, which is the KKT-type first-order condition for an $\varepsilon$-Pareto stationary point. If the zero vector is not in a neighborhood of that convex hull, choosing the maximizing $\mathbf{w}$ yields a common-descent direction that contradicts the no-better-weight condition.

PRISM's outer loop can be written explicitly as a gradient update over a weighted combination of deficiencies:

$$\theta^{(t+1)} = \theta^{(t)} - \eta \sum_{i=1}^n w_i\big(\theta^{(t)}\big)\, \nabla a_i\big(\theta^{(t)}\big), \qquad \mathbf{a}^{(t+1)} = \mathbf{a}\Big(\theta^{(t+1)}\Big), \tag{7}$$

We monitor $\mathbf{a}^{(t)}$ with a coordinatewise tolerance $\varepsilon$ and patience $R$. If every coordinate fails to decrease by more than $\varepsilon$ for $R$ consecutive checks, training stops. Because $W(\cdot)$ is recomputed from current diagnostics, at the stopping iterate $\theta^{(t^\star)}$ the first-order improvement obtainable by any feasible weight $\mathbf{w}$ is bounded by $\varepsilon$:

$$\frac{d}{d\eta}\, a_i\Big(\theta^{(t^\star)} + \eta\, d(\theta^{(t^\star)}; \mathbf{w})\Big)\bigg|_{\eta=0} = -\nabla a_i(\theta^{(t^\star)})^\top \sum_j w_j \nabla a_j(\theta^{(t^\star)}) \ \leq\ \varepsilon.$$

Under standard smoothness assumptions and small step size, the second-order remainder yields

$$a_i\Big(\theta^{(t^\star)} + \eta\, d(\theta^{(t^\star)}; \mathbf{w})\Big) \geq a_i\Big(\theta^{(t^\star)}\Big) - \eta\,\varepsilon - O(\eta^2).$$

Hence there is no sampling weight that, after one update, produces a uniformly larger-than-$\varepsilon$ improvement across all objectives. In words, PRISM alternates training on samples drawn by weights induced from diagnostics and recomputing those weights from fresh diagnostics; it terminates exactly when no further reweighting can generate a common-descent direction of practical significance. The returned parameter $\theta^{(t^\star)}$ is therefore an $\varepsilon$-Pareto stationary compromise for the vector of deficiencies, achieved without enumerating models along the frontier but by exhausting all feasible common-descent directions induced by sampling weights.

# 4 EXPERIMENTS

## 4.1 EXPERIMENTAL SETUP

We follow the same experimental settings as the ScoreFlow baseline to ensure a fair comparison. In all experiments, we use the LLaMA 3.1-8B Instruct model (Grattafiori et al., 2024) as the base LLM. Training is performed on two NVIDIA A6000 GPUs, and we adopt the same learning rate and LoRA fine-tuning configuration reported by ScoreFlow so that optimizer, rank, learning-rate schedule, and batch sizing are matched.

We evaluate on six benchmarks across three domains: QA (HotpotQA (Yang et al., 2018), DROP (Dua et al., 2019)), coding (HumanEval (Chen, 2021), MBPP (Austin et al., 2021)), and math reasoning (GSM8K (Cobbe et al., 2021), MATH (Hong et al., 2025)). For each dataset, we sample

multiple candidate plans and form preference pairs by comparing their outcomes. Plan generation follows the dataset's fixed step types from ScoreFlow. In the single-objective setting we allow only the four task-appropriate types; in the multi-objective setting we enlarge the action space with off-domain step types to expose robustness and "wrong" actions.

Plan scoring is automated. A GPT-4o-mini (OpenAI, 2024) based executor runs each plan to produce a final answer and an execution trace. Accuracy is computed as exact-match on HotpotQA, DROP, GSM8K, and MATH, and by canonical unit tests on HumanEval and MBPP. Efficiency and robustness are derived from the trace: wall-clock reasoning time (or its proxy latency), the number of reasoning steps, and the count of task-irrelevant steps. For every input–plan pair $(x, y)$, we record a four-dimensional objective vector $\mathbf{O}(x, y) = (\text{Acc}, -\text{Time}, -\text{Step}, -\text{Wrong})$ so that larger is better along all coordinates. We do not collapse these objectives into a single scalar; PRISM consumes the vector directly and uses objective-gap directions to orient preference pairs and drive DPO updates as described in the Methodology.

Training uses Direct Preference Optimization on the collected plan-preference data. We compare PRISM against standard prompting/reasoning baselines (IO, CoT), self-refinement (Self-Refine), adaptive planning (ADAS, Aflow), and the ScoreFlow approach. For multi-objective alignment we include MORLHF, MO-ODPO, and MODPO. Evaluation reports final solution accuracy (higher is better) and three efficiency/robustness metrics—time, steps, and wrong-step count (lower is better for all). Accuracy is treated as the primary objective, avoiding wrong steps is treated as a hard constraint, and minimizing time and steps are secondary objectives for efficiency.

## 4.2 SINGLE-OBJECTIVE RESULTS

With the planner restricted to task-appropriate step types and a single objective, PRISM remains competitive across question answering, coding, and math reasoning. Under identical settings, PRISM leads on 5 of 6 benchmarks and raises the cross-task macro average by about 1.7 percentage points relative to the strongest baseline, indicating a consistent lift without task-specific tuning. Qualitatively, the largest gains occur where candidate solutions contain many plausible but misleading traces (e.g., program synthesis and multi-step derivations), suggesting that preference-based plan selection helps the model favor structurally coherent reasoning over superficially similar alternatives. On datasets already near saturation, performance is effectively tied, reflecting limited headroom rather than a limitation of the approach. Overall, Table 1 shows that PRISM delivers solid single-objective accuracy improvements purely from preference optimization, independent of action-space expansion or auxiliary signals.

Table 1: Comparison between PRISM and other baselines in single objective optimization

| Method | Question Answering | | Coding | | Math Reasoning | | Average |
| --- | --- | --- | --- | --- | --- | --- | --- |
| | HotpotQA | DROP | HumanEval | MBPP | GSM8K | MATH | |
| IO | 73.6% | 81.6% | 90.1% | 69.5% | 89.1% | 52.2% | 76.0% |
| COT | 73.4% | 83.2% | 91.6% | 70.4% | 88.3% | 53.4% | 76.7% |
| Self Refine | 73.6% | 82.5% | 91.1% | 70.0% | 87.5% | 50.0% | 75.8% |
| ADAS | 78.5% | 81.3% | 88.8% | 68.7% | 90.5% | 51.7% | 76.6% |
| Aflow | 77.9% | 83.5% | 92.9% | 82.9% | 90.8% | 55.8% | 80.6% |
| Scoreflow | 86.0% | 86.2% | 95.9% | 84.7% | **94.6%** | 64.4% | 85.3% |
| PRISM | **87.4%** | **88.9%** | **96.8%** | **87.6%** | 94.1% | **67.6%** | **87.0%** |

## 4.3 MULTI-OBJECTIVE RESULTS

To approximate real deployments, we expand each task's action space by injecting distractor step types that belong to other domains: for example, question answering receives coding- and math-specific operators; coding receives QA and math operators; math reasoning receives QA and coding operators. This forces the planner to navigate a larger space and exposes whether it can actively avoid irrelevant actions. We record four objectives per executed plan: $\text{Acc}$ for evaluator-judged correctness, $\text{Time}$ for executor wall-clock latency, $\text{Step}$ for the number of reasoning steps, and $\text{Wrong}$ for the count of off-domain steps taken. Accuracy is treated as the primary objective, $\text{Wrong}$ as a hard constraint, and $\text{Time}$ and $\text{Step}$ as efficiency objectives.

Table 2 shows that PRISM achieves the strongest overall trade-off. It leads accuracy on most tasks while keeping wrong-step usage essentially zero, indicating that Pareto-guided orientation reliably avoids distractors. In terms of efficiency, PRISM also attains the lowest average latency ($0.8573$ s versus $0.8867$ s for MODPO and $0.9355$ s for MO-ODPO) and the shortest plans on average ($2.8038$ reasoning steps versus $3.5141$ and $3.8292$, respectively). Crucially, this is achieved without sacrificing correctness: PRISM reaches a cross-task accuracy of $0.8479$ compared to $0.8249$ (MODPO) and $0.8141$ (MO-ODPO).

Beyond runtime efficiency, PRISM is also compute-efficient during training. The GPU-Hour values reported in Table 2 already include the *full* computational footprint of each method. For PRISM, the total of $1.8475$ GPU-Hours consists of both standard DPO updates ($1.4167$ GPU-Hours) and the additional deficiency-diagnostic computations on golden comparisons ($0.4308$ GPU-Hours), so diagnostics account for about $23.3\%$ of the overall cost. Despite this overhead, PRISM remains more efficient than Panacea ($-6.33\%$ GPU-H), MO-ODPO ($-2.31\%$), and MODPO ($-92.16\%$), all of which rely on reinforcement learning or multiple runs under different objective weights. PRISM instead converges in a single, fully offline training run by dynamically steering updates toward Pareto-improving directions.

Table 2: Comparison between PRISM and other baselines in multi-objective optimization processing

| Method | Object | Question Answering | | Coding | | Math Reasoning | | Average | GPU-H |
|---|---|---|---|---|---|---|---|---|---|
| | | HotpotQA | DROP | HumanEval | MBPP | GSM8K | MATH | | |
| Scoreflow | Acc% | 0.7842 | 0.7854 | 0.9137 | 0.7388 | 0.8784 | 0.5404 | 0.7735 | 1.4667 |
| | Time(s) | 0.7413 | 0.7549 | 2.0504 | 0.7837 | 0.7027 | 0.8021 | 0.9725 | |
| | #Step | 3.3147 | 2.5641 | 6.4512 | 4.2420 | 10.2000 | 6.5214 | 5.5489 | |
| | #Wrong | 1.7150 | 1.5980 | 1.4329 | 1.3860 | 1.5360 | 0.9080 | 1.4293 | |
| MORLHF | Acc% | 0.7215 | 0.7461 | 0.8223 | 0.6947 | 0.8169 | 0.4810 | 0.7138 | **1.3534** |
| | Time(s) | 0.8006 | 0.8002 | 2.2554 | 0.8386 | 0.7378 | 0.8743 | 1.0512 | |
| | #Step | 3.5467 | 2.6923 | 7.0963 | 4.5814 | 11.1180 | 6.9112 | 5.9910 | |
| | #Wrong | 1.8865 | 1.7099 | 1.5619 | 1.4553 | 1.6589 | 0.9625 | 1.5392 | |
| Panacea | Acc% | 0.8123 | 0.8037 | 0.9315 | 0.8228 | 0.8921 | 0.6034 | 0.8110 | 1.9723 |
| | Time(s) | 0.7394 | 0.7316 | 2.2147 | 0.7613 | 0.5839 | 0.6528 | 0.9473 | |
| | #Step | 2.7045 | 2.1372 | 4.1563 | 3.2189 | 6.0314 | 5.0876 | 3.8893 | |
| | #Wrong | 1.1935 | 1.1472 | 1.4029 | 1.1287 | 1.5974 | 0.6931 | 1.3771 | |
| MO-ODPO | Acc% | 0.8194 | 0.7933 | 0.9320 | 0.8311 | 0.9004 | 0.6083 | 0.8141 | 1.8911 |
| | Time(s) | 0.7353 | 0.7105 | 2.2493 | 0.7378 | **0.5510** | 0.6291 | 0.9355 | |
| | #Step | 2.8352 | 2.2560 | 3.7622 | 2.9840 | 6.3540 | 4.7840 | 3.8292 | |
| | #Wrong | 0.1872 | 0.1240 | 0.3659 | 0.1140 | 0.8640 | 0.4800 | 0.3559 | |
| MODPO | Acc% | 0.8239 | 0.8453 | **0.9571** | 0.8132 | 0.8964 | 0.6133 | 0.8249 | 23.5716 |
| | Time(s) | 0.7178 | 0.7249 | 1.8732 | 0.7630 | 0.6246 | 0.6166 | 0.8867 | |
| | #Step | 2.3738 | 1.8501 | 4.7317 | 3.0140 | 5.0368 | 4.0780 | 3.5141 | |
| | #Wrong | 0.0276 | **0.0000** | 0.0932 | 0.1074 | 0.0894 | 0.0629 | 0.0634 | |
| PRISM | Acc% | **0.8692** | **0.8773** | 0.9489 | **0.8389** | **0.9264** | **0.6269** | **0.8479** | 1.8475 |
| | Time(s) | **0.7136** | **0.6928** | **1.8432** | 0.7335 | 0.5619 | **0.5990** | **0.8573** | |
| | #Step | **1.9389** | **1.8120** | **3.1281** | **2.5880** | **3.3540** | **4.0020** | **2.8038** | |
| | #Wrong | **0.0000** | **0.0000** | **0.0006** | **0.0024** | **0.0000** | **0.0376** | **0.0068** | |

Overall, these results show that once distractor actions are introduced, preference learning over objective-gap directions is critical. By combining golden diagnostics with adaptive weighting and cosine-oriented sampling, PRISM consistently stays on a favorable region of the Pareto surface—high accuracy, few or no wrong actions, and lower time and step costs—outperforming alternative multi-objective DPO approaches on the combined efficiency metrics while maintaining strong correctness.

## 4.4 Weight Evolution and Exploration under the Pareto Frontier

As shown in Fig. 2, we visualise the evolution of PRISM's sampling weights throughout training while probing the Pareto front for each dataset. Since the four objective weights satisfy $w_{\text{Acc}} + w_{\text{Time}} + w_{\text{Step}} + w_{\text{Wrong}} = 1$, we employ a two-dimensional simplex (ternary) projection to display their joint behaviour. The three axes correspond to the weights on Accuracy, Time, and Step, while the weight on Wrong is implicitly given by $w_{\text{Wrong}} = 1 - (w_{\text{Acc}} + w_{\text{Time}} + w_{\text{Step}})$.

Each polyline connects weight-change checkpoints across epochs. Each marker corresponds to one outer iteration of PRISM, where the sampling weights are re-estimated from current deficiency diagnostics. Therefore, every point on the polyline represents an actual weight update, and each polyline traces how the weights evolve over successive epochs.

Importantly, the trajectories cover a wide region of the simplex instead of converging immediately to a local corner. This indicates that PRISM systematically explores the objective space rather than collapsing early to a fixed trade-off. The tick marks along the simplex edges show the magnitude of each objective's weight, and the movement across different areas of the triangle demonstrates that attention indeed shifts among objectives during training.

Because the weights are recomputed from golden-comparison diagnostics at every outer iteration, they naturally shift toward objectives exhibiting higher deficiency (i.e., currently under-optimised). Consequently, the sampler prioritises preference pairs whose objective-gap directions are most aligned with the present deficiencies, increasing the scalarised improvement per update and guiding the policy toward balanced solutions.

This adaptive evolution ultimately stabilises in ridge regions of the simplex, indicating convergence to a Pareto-consistent operating point where no objective can be further improved without degrading another.

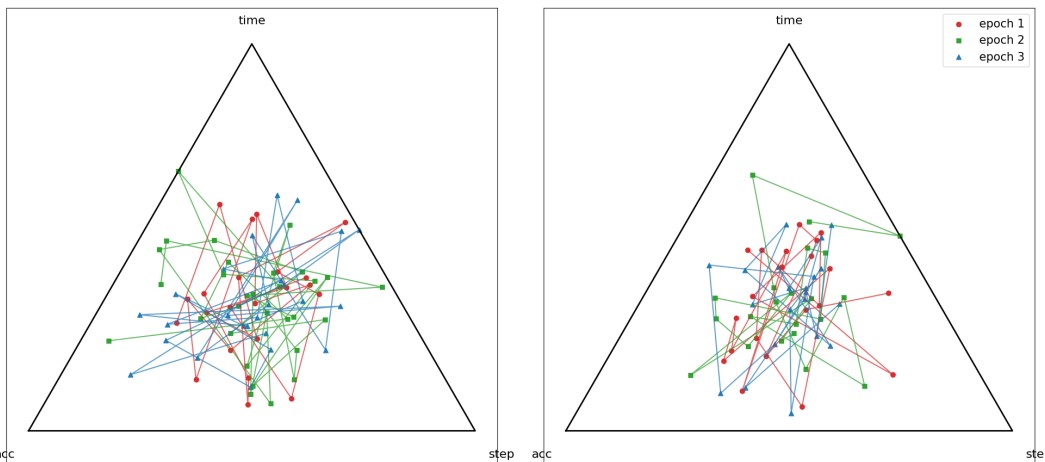

Figure 2: Weight exploration graph based on Pareto front under different datasets. Trajectories show the evolution of the four simplex-weights (which sum to one) across epochs, visualised via a triangular (simplex) plot.

## 4.5 ABLATION STUDY

Table 3 summarizes ablations across six datasets. Removing golden comparisons strips away single-objective diagnostics and substantially degrades all metrics, confirming their necessity. Using only loss or only gradient norm to build the deficiency signal is insufficient: the former preserves accuracy but hurts efficiency, whereas the latter collapses performance across the board. On sampling, Threshold and Top-K remain cosine based they select pairs by applying a hard cutoff or cardinality cap on the cosine alignment used by PRISM and thus retain reasonable accuracy but allow more wrong steps than our proportional scheme. In contrast, the Euclidean variant ranks pairs by distance rather than directional agreement, which conflicts with the goal of steering updates toward currently weak objectives and yields weaker results. Finally, removing the warm-start slows optimization and produces longer plans, indicating that a brief stabilization phase helps avoid poor orientations. Overall, only the full configuration achieves the desired balance of accuracy, time, step count, and near-zero wrong actions.

## 4.6 CLOSED-MODEL BASELINES UNDER HIDDEN CONSTRAINTS

State-of-the-art closed models such as Gemini 2.5 Pro and GPT-5 already achieve near-saturation success on many public benchmarks; nevertheless, in enterprise and safety-critical deployments the ability to fine-tune on proprietary data and encode domain-specific preferences remains decisive. To reflect this reality, we evaluate under hidden constraints that are invisible to the policy but enforced in the reward (for example, answers produced with fewer than two reasoning steps are scored as incorrect). The full list of such instructions and compliance rules is provided in Appendix 5.

Table 3: Component ablations on three representative datasets (HotpotQA, MBPP, GSM8K), averaged over 5 seeds. Higher is better for **Acc**; lower is better for **Time/Step/Wrong**.

| Category | Variant | Acc%↑ | Time(s)↓ | #Step↓ | #Wrong↓ |
|---|---|---|---|---|---|
| Full | Ours | **0.8479** | **0.8573** | **2.8038** | **0.0068** |
| Architecture | No-Golden | 0.7815 | 1.1462 | 5.9620 | 1.5125 |
| | No-Warm | 0.8192 | 0.8924 | 3.2541 | 0.2196 |
| Parameter | Only-Loss ($\gamma$=1) | 0.8340 | 0.8760 | 2.9200 | 0.1082 |
| | Only-Grad ($\gamma$=0) | 0.5287 | 2.8894 | 6.2415 | 3.0285 |
| Sampling | Euclidean | 0.8112 | 1.1861 | 3.5118 | 1.2724 |
| | Threshold | 0.8221 | 0.9642 | 3.1826 | 0.1123 |
| | Top-K | 0.8343 | 0.8810 | 3.0217 | 0.1101 |
| | Random | 0.6293 | 2.4461 | 5.4941 | 2.9451 |

Evaluation protocol. Gemini 2.5 Pro and GPT-5 are invoked via their APIs and serve as both generator and executor within their native toolchains. All systems face the same hidden constraints and expanded action spaces with cross-domain distractor steps, and we report accuracy, time, step count, and off-domain usage.

Table 4: The latest model performance under all instruction errors (simulating GPT without learning relevant knowledge)

| Method | Object | Question Answering | | Coding | | Math Reasoning | | Average |
|---|---|---|---|---|---|---|---|---|
| | | HotpotQA | DROP | HumanEval | MBPP | GSM8K | MATH | |
| Gemini 2.5 Pro | Acc% | 0.4187 | 0.4072 | 0.4668 | 0.4573 | 0.3925 | 0.3861 | 0.4214 |
| | Time(s) | 0.7924 | 0.8136 | **0.7631** | **0.7548** | 0.8063 | **0.8297** | 0.7933 |
| | #Step | **2.1846** | 2.7134 | **2.3319** | 2.2712 | 2.6945 | 2.8538 | 2.5099 |
| | #Wrong | 2.3547 | 2.4271 | 2.1836 | 2.2569 | 2.3924 | 2.4418 | 2.3428 |
| GPT5 | Acc% | 0.4871 | 0.4746 | 0.4028 | 0.4147 | 0.4985 | 0.4693 | 0.4578 |
| | Time(s) | **0.7017** | **0.7198** | 0.8272 | 0.8046 | **0.6841** | 0.8394 | 0.7628 |
| | #Step | 2.2791 | **2.4637** | 2.7886 | 2.7415 | 2.1932 | 2.6124 | **2.5131** |
| | #Wrong | 2.1189 | 2.0813 | 2.3347 | 2.2711 | 2.1538 | 2.2873 | 2.2079 |
| PRISM | Acc% | **0.7794** | **0.7751** | **0.8492** | **0.7447** | **0.8351** | **0.5353** | **0.7531** |
| | Time(s) | 1.6941 | 1.7032 | 2.9027 | 1.6838 | 1.5526 | 1.6294 | 1.8610 |
| | #Step | 3.9492 | 3.8927 | 5.1674 | 4.5783 | 5.5146 | 5.9228 | 4.8375 |
| | #Wrong | **0.0107** | **0.0099** | **0.0114** | **0.0132** | **0.0108** | **0.0487** | **0.0175** |

As shown in Table 4, PRISM achieves markedly higher accuracy under hidden constraints (about 0.75 on average) while keeping off-domain actions near zero, whereas GPT-5 and Gemini 2.5 Pro average roughly 0.46 and 0.42 with multiple off-domain steps. This reflects the advantage of open-weight fine-tuning, which aligns planning with enterprise policies that are not visible at inference. Although PRISM incurs higher latency and longer traces, this trade-off is acceptable in regulated settings where controllability and auditability outweigh raw throughput.

## 5 CONCLUSION

We presented PRISM, a multi-objective preference optimization framework for LLM planning that combines golden comparisons, loss–gradient fusion for adaptive weighting, and Pareto-guided sampling to orient updates by objective gap directions. On six datasets spanning QA, coding, and math reasoning, PRISM improves accuracy while nearly eliminating wrong steps and reducing time and step costs, outperforming ScoreFlow, MORLHF, MO-ODPO, and MODPO. In production at an Australian real-estate company, PRISM serves as the workflow planner and delivers at least a 34% gain in aggregate workflow quality, demonstrating practical benefits under real constraints and domain-specific preferences.

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

## APPENDIX A

In this work, large language models (LLMs) were used *only* for light language polishing (grammar and wording refinement) after the authors had completed the technical content. No LLMs were used to generate or edit scientific claims, ideas, experimental designs, code, data, figures, tables, derivations, proofs, or citations. All analyses, methods, experiments, and results were conceived, implemented, and validated solely by the authors. All references were selected and checked by the authors, and no AI-generated citations were introduced. No confidential, proprietary, or personally identifiable information was provided to any LLM or external service.

This disclosure aligns with current community and publisher guidance that AI tools must not be credited as authors and that any AI assistance should be transparently reported while accountability remains with the human authors.

This work does not involve human subjects research, behavioral intervention, or collection of personally identifiable information, and thus does not require Institutional Review Board approval. All datasets used are publicly available under their respective licenses; we follow the usage terms and provide clear attribution. When constructing preference data and golden comparisons, we filter prompts and generated content to avoid unsafe, discriminatory, or privacy-violating outputs, and we exclude tasks that solicit harmful actions. Closed-source APIs are queried only for evaluation where permitted by their terms of service; no proprietary content is redistributed. For deployed use, the method is intended to reduce off-domain and unsafe actions by explicitly penalizing such steps. We encourage downstream users to conduct additional safety evaluations appropriate to their domain before deployment.

We provide code, configuration files, and scripts to reproduce all results, including data preprocessing, plan generation, executor evaluation, preference-pair construction, golden-comparison selection, training, and inference. The repository contains: exact prompts for generator and executor, objective definitions, hidden-constraint rules, and scoring code; training hyperparameters (optimizer, learning rate schedule, LoRA settings, batch sizes, number of updates), random seeds, and checkpointing routines; environment specifications with package versions and hardware requirements. To address reliance on external evaluators, we release cached evaluator judgments for all runs reported and include an open-weight evaluator that mirrors the decision rules used in the paper. We fix seeds for sampling, shuffling, and model initialization, and we document any remaining nondeterminism due to GPU kernels. For each figure and table, we include a one-click script that regenerates the artifact from released logs and checkpoints. We report training tokens, wall-clock time, and compute hardware to facilitate cost and environmental impact estimation; an optional script is included to log energy usage during reruns.

## APPENDIX B    HIDDEN CONSTRAINTS AND REWARD SHAPING

This appendix lists the hidden constraints used to simulate enterprise policies and domain preferences. These rules are invisible to the policy during inference and are enforced only in the reward computation. They induce cooperation and conflict among objectives by tying accuracy to minimal reasoning quality, latency budgets, and correct use of domain-appropriate steps. We use the same notation as in the main text: $\mathrm{Acc} \in \{0, 1\}$ is evaluator-judged correctness, $\mathrm{Time}$ is executor wall-clock latency, $\mathrm{Step}$ is the number of reasoning steps, and $\mathrm{Wrong}$ counts off-domain actions. Unless otherwise stated, updates are applied in the order shown.

Table 5: Part of hidden constraints and scoring effects

| Tension | Rule (formal condition) | Score update on ($\mathrm{Acc}, \mathrm{Time}, \mathrm{Step}, \mathrm{Wrong}$) |
|---|---|---|
| Acc vs Step (cooperation) | Minimum depth: if $\mathrm{Step} < 2$ | Set $\mathrm{Acc} \leftarrow 0.5 \cdot \mathrm{Acc}$. |
| Time vs Wrong (conflict) | Latency budget: if $\mathrm{Time} > 8$ | Set $\mathrm{Wrong} \leftarrow \mathrm{Wrong} + 1$. |
| Acc vs Wrong (conflict) | Off-domain hardening: if $\mathrm{Wrong} > 0$ and $\mathrm{Acc} = 1$ | Set $\mathrm{Acc} \leftarrow 0.8$. |
| Step vs Time (conflict) | Redundancy suppression: for each repeated identical step beyond the first | Set $\mathrm{Time} \leftarrow \mathrm{Time} + 0.2$. |
| Acc vs Wrong (conflict) | Tool/step misuse: using a code-only operator in QA/proof, or a proof-only operator in coding/QA | Set $\mathrm{Wrong} \leftarrow \mathrm{Wrong} + 1$. |
| Acc vs Time (cooperation) | Verification preference: if the last step includes an explicit check/verify and $\mathrm{Wrong} = 0$ | Set $\mathrm{Acc} \leftarrow \min(1, \mathrm{Acc} + 0.05)$ and $\mathrm{Time} \leftarrow \mathrm{Time} + 0.1$. |
| Step vs Time (conflict) | Budgeted length: if $\mathrm{Step} > 12$ | Set $\mathrm{Time} \leftarrow \mathrm{Time} + 1$. |
| Time vs Wrong (conflict) | External-call quota: if the number of executor API/tool calls is greater than four | Set $\mathrm{Time} \leftarrow \mathrm{Time} + 0.5$ and $\mathrm{Wrong} \leftarrow \mathrm{Wrong} + 1$. |
| Acc vs Time (conflict) | Premature finalization: if an answer is emitted and later revised within the same plan | Set $\mathrm{Acc} \leftarrow 0.9 \cdot \mathrm{Acc}$ and $\mathrm{Time} \leftarrow \mathrm{Time} + 0.2$. |
| Acc vs Wrong (conflict) | Irrecoverable contradiction: final answer contradicts evidence in the trace | Set $\mathrm{Wrong} \leftarrow \mathrm{Wrong} + 1$ (no direct change to $\mathrm{Acc}$ beyond evaluator judgment). |

**Rationale and interactions.**    Depth (minimum steps) ties correctness to minimal reasoning quality, discouraging one-step guessing while allowing concise solutions that meet the floor. Latency and length budgets cooperate with efficiency but may conflict with accuracy when problems need longer derivations. Off-domain penalties harden compliance, aligning with $\mathrm{Wrong}$ but potentially reducing $\mathrm{Acc}$ on otherwise-correct traces that relied on inappropriate operators. Redundancy suppression translates loops into small time taxes, making compact plans more attractive. A mild verification bonus trades a small latency increase for reliability. External-call and premature-finalization rules reflect enterprise quotas and discourage answer churn. Evidence–answer contradictions are recorded as compliance risk via $\mathrm{Wrong}$, decoupled from the evaluator's binary correctness.

**Application notes.**    All constraints are invisible to the policy; they modify only the measured objective vector before preference construction. In PRISM, the shaped objectives yield gap vectors that interact with the adaptive weight direction, producing sampling that favors plans respecting enterprise constraints without exposing these rules at inference time.

## APPENDIX C   HIDDEN CONSTRAINTS AND REWARD SHAPING

Figure 3 visualizes, for each dataset, how the planner's best plan varies under different multi-objective weights and how PRISM tracks the Pareto surface. The triangular base represents the simplex of weight allocations for three displayed coordinates $(w_1, w_2, w_3)$, with the fourth weight recovered by $w_4 = 1 - w_1 - w_2 - w_3$. Each point on the base is thus a specific trade-off over the four objectives. For a given weight vector $w$, we evaluate all candidate plans and select the plan that maximizes the weighted score; the vertical bar at that base location shows the resulting best score, and the translucent surface is the upper envelope of these best scores over the entire weight simplex.

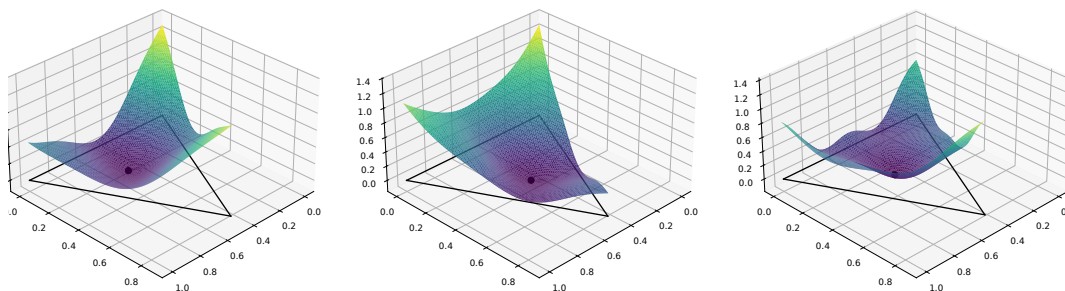

Figure 3: Pareto front of HotpotQA, HumanEval, and GSM8K during training

To make scores comparable across objectives, we keep accuracy in its native scale and apply a reciprocal transform to efficiency and robustness metrics so that "larger is better" for all coordinates. Concretely, accuracy is used as is, while time, step, and wrong are mapped to $\tilde{t} = 1/(1 + \text{Time})$, $\tilde{s} = 1/(1 + \text{Step})$, and $\tilde{w} = 1/(1 + \text{Wrong})$, which are monotone in the desirable direction and bounded in $(0, 1]$. The overall bar height at a weight $w$ is therefore consistent across datasets and emphasizes plans that jointly reduce latency, length, and off-domain actions without sacrificing correctness.

The isolated markers on the surface denote the Pareto-optimal plans that PRISM visits during training under its current adaptive weights. As training proceeds, points migrate toward ridges of the surface where no objective can be improved without degrading another, indicating convergence toward balanced compromises. On HotpotQA and MBPP the ridges are elongated along the efficiency axes, reflecting sizeable headroom in time and step reduction once wrong actions are suppressed. On GSM8K the surface is flatter near accuracy-dominant corners, consistent with saturation under single-objective constraints and smaller marginal gains from further efficiency pressure. Overall, the figure illustrates that PRISM's updates concentrate on regions with steep directional improvement and then stabilize along the Pareto front, yielding solutions that remain competitive across a broad range of weightings rather than overfitting to a single operating point.

## APPENDIX D    DERIVATION OF THE STOPPING CONDITION

Recall that $d(\theta; w) = -G(\theta)w$, where $G(\theta) = [g_1(\theta), \ldots, g_n(\theta)]$ collects the gradient vectors of all objective deficiencies. For a small step size $\eta > 0$, a descent of $a_i(\theta)$ along $d(\theta; w)$ occurs if

$$\frac{d}{d\eta} a_i(\theta + \eta d(\theta; w))\Big|_{\eta=0} = -\nabla a_i(\theta)^\top d(\theta; w) > 0.$$

If there exists a weight vector $w \in \Delta^{n-1}$ such that this quantity is positive for all $i$, then $d(\theta; w)$ provides a common-descent direction that simultaneously reduces all deficiencies. Conversely, if for all feasible $w$,

$$\max_i \big\{ -\nabla a_i(\theta)^\top d(\theta; w) \big\} \leq \varepsilon,$$

no such direction yields an improvement beyond $\varepsilon$. This is equivalent to stating that the zero vector lies (up to $\varepsilon$) in the convex hull of $\{g_i(\theta)\}_{i=1}^n$, which is the first-order KKT condition for an $\varepsilon$-Pareto stationary point. This justifies terminating the outer loop when the above condition is met.

# APPENDIX E    HYPERPARAMETER SETTINGS AND DEFICIENCY SIGNAL CONFIGURATION

Table 6 summarizes all hyperparameters used in PRISM. Unless otherwise stated, the same configuration is applied across all six benchmarks. Values were selected based on preliminary tuning on the validation split of the ScoreFlow setup, followed by cross-checking consistency across seeds.

Table 6: Hyperparameter settings used in all experiments.

| Parameter | Value | Description |
|---|---|---|
| $\gamma$ | **0.95** | Weight of loss in deficiency score $a_i = \gamma\bar{\ell}_i + (1-\gamma)\bar{g}_i$ |
| $\beta$ | **0.1** | Cosine sampling sharpness in $q(x; y, y') \propto \lvert s(x; y, y')\rvert^\beta$ |
| $M$ | 32 | Golden comparisons per objective per iteration (diagnostics only) |
| $\lambda$ | Linear decay $1 \to 0$ | Curriculum bias on primary vs hardness objectives |
| $\tau$ | 0.8 | Softmax temperature in weight normalization |
| $\epsilon$ | $10^{-3}$ | Pareto stationarity tolerance for stop condition |
| $R$ | 3 | Patience for early stopping check |
| Warm-up (random comparisons) | 100 pairs | Stabilization of weight computation |
| Optimizer | AdamW | Same config as ScoreFlow |
| Learning rate | $1 \times 10^{-4}$ | Inherited from ScoreFlow setup |
| LoRA rank $r$ | 64 | Adapter dimension |
| Batch size | 128 | Preference pair training |
| Max epochs | 8 | Outer-loop training |

**Rationale for $\gamma$ and $\beta$.** The deficiency signal combines the average loss $\bar{\ell}_i$ and gradient norm $\bar{g}_i$ per objective:

$$a_i = \gamma\bar{\ell}_i + (1-\gamma)\bar{g}_i.$$

The two extreme cases appear as ablations in Table 3: **Only-Loss** ($\gamma = 1$) preserves accuracy but degrades efficiency and robustness, while **Only-Grad** ($\gamma = 0$) collapses performance across all metrics. We therefore adopt a high-but-non-degenerate value $\gamma = 0.95$ that retains strong sensitivity to error frequency while preserving corrective difficulty.

The cosine-based sampling probability

$$q(x; y, y') \propto \lvert s(x; y, y')\rvert^\beta$$

uses a small value $\beta = 0.1$ to avoid over-concentration while still preferring weight-aligned comparisons. Ablations (Euclidean, Threshold, Top-$K$) confirm robustness w.r.t sampling choice, with PRISM performing best when mild sharpening is used.

**Sensitivity.** A small-scale tuning over $\gamma \in \{0.5, 0.8, 0.95, 1.0\}$ showed that $\gamma = 0.95$ consistently improves multi-objective trade-off by up to $\sim 1.2\%$ in accuracy and reduces wrong-step usage compared to $\gamma = 0.8$ or $\gamma = 1.0$. $\beta = 0.1$ yielded the lowest variance across seeds. Further adaptive tuning is left as future work.

