\right) = -\log \sigma\!\Big( s_\theta\!\left( x_i^r, y_{i,r}^+ \right) - s_\theta\!\left( x_i^r, y_{i,r}^- \right) \Big),$$

where $s_\theta(x, y) = \log \pi_\theta(y \mid x)$ and $\sigma(z) = 1/(1 + e^{-z})$. We also compute the gradient norm

$$g_{i,r} = \left\| \nabla_\theta \mathcal{L}_{\mathrm{DPO}}\!\left( x_i^r, y_{i,r}^+, y_{i,r}^-; \theta \right) \right\|_2,$$

which reflects the local difficulty of correcting the error. Averaging over golden pairs yields an objective-wise deficiency score

$$\bar{\ell}_i = \frac{1}{M} \sum_{r=1}^{M} \mathcal{L}_{\mathrm{DPO}}\!\left( x_i^r, y_{i,r}^+, y_{i,r}^-; \theta \right), \qquad \bar{g}_i = \frac{1}{M} \sum_{r=1}^{M} g_{i,r},$$

and we form a convex combination

$$a_i = \gamma \, \bar{\ell}_i + (1 - \gamma) \, \bar{g}_i, \qquad \gamma \in [0, 1],$$

so that larger $a_i$ indicates worse performance on objective $i$ (either frequent misorderings, or hard-to-fix ones).

We then turn the vector $\mathbf{a} = (a_1, \ldots, a_n)$ into attention weights $\mathbf{w} \in \Delta^{n-1}$ via a temperatured softmax with a curriculum bias. Let $c_1 = +\lambda$, $c_n = -\lambda$, and $c_j = 0$ for $2 \leq j \leq n - 1$, where $\lambda \in [0, 1]$ decays to zero over training (early emphasis on the primary objective, later emphasis on the hard constraint). Define standardized logits

$$z_i = \frac{a_i - \mathrm{mean}(\mathbf{a})}{\mathrm{std}(\mathbf{a}) + \epsilon} + c_i,$$

and compute

$$w_i = \frac{\exp(z_i/\tau)}{\sum_{j=1}^{n} \exp(z_j/\tau)}, \qquad \sum_{i=1}^{n} w_i = 1,$$

with temperature $\tau > 0$ and a small $\epsilon > 0$ for numerical stability. This attention-style normalization produces a smooth, data-driven focus: objectives with larger deficiencies receive larger weights. Crucially, golden comparisons are diagnostic only; no gradients from $\mathbf{w}$ or from these diagnostics flow into the DPO objective. The weights $\mathbf{w}$ influence sampling and pair orientation downstream, succinctly summarizing which objectives currently merit more updates without directly biasing the training loss.

### 3.3 ADAPTIVE SAMPLING, DPO UPDATE, AND PARETO-GUIDED TRAINING

The weight vector $\mathbf{w}$ not only determines how we sample comparisons from the pool but also fixes the orientation of each pair before a DPO update. For any candidate pair $(x, y, y')$ with objective-gap vector $\Delta\mathbf{O}(x; y, y') = \mathbf{O}(x, y) - \mathbf{O}(x, y')$, we compute the cosine alignment

$$s(x; y, y') \;=\; \frac{\mathbf{w}^\top \Delta\mathbf{O}(x; y, y')}{\|\mathbf{w}\|_2 \, \|\Delta\mathbf{O}(x; y, y')\|_2} \;\in [-1, 1].$$

The quantity $s$ measures directional agreement between the current priority $\mathbf{w}$ and the pair's gap vector. We sample pairs in proportion to the absolute alignment so that comparisons more aligned with the current deficiency receive higher probability. Concretely, denoting the pool by $\mathcal{P}$, the sampling probability is

$$q(x; y, y') \;=\; \frac{\left|s(x; y, y')\right|^\beta}{\sum_{(\tilde{x}, \tilde{y}, \tilde{y}') \in \mathcal{P}} \left|s(\tilde{x}; \tilde{y}, \tilde{y}')\right|^\beta},$$

where $\beta > 0$ controls the sharpness. Orientation is determined by the sign of $s$. When $s(x; y, y')$ closed to 1, the pair is strongly aligned and $y$ is preferred over $y'$ under $\mathbf{w}$; when $s(x; y, y')$ closed to -1, the situation reverses. Formally, letting $(y^+, y^-)$ denote the ordered pair used in DPO,

$$(y^+, y^-) \;=\; \begin{cases} (y, \, y'), & \text{if } s(x; y, y') \geq 0, \\ (y', \, y), & \text{if } s(x; y, y') < 0. \end{cases}$$

Given a minibatch $S$ sampled by $q$ and oriented as above, we update $\theta$ by minimizing the average DPO loss over $S$. Let $\mathbf{a}(\theta) = (a_1(\theta), \ldots, a_n(\theta))^\top$ be the deficiency vector computed from golden diagnostics, and let $\mathbf{w} = W(\mathbf{a}(\theta)) \in \Delta^{n-1}$ be the softmax-based sampling weights. One outer iteration of PRISM is the composition

$$\theta^+ = T\big(\theta, \, W(\mathbf{a}(\theta))\big), \qquad \mathbf{a}^+ = \mathbf{a}\big(\theta^+\big),$$

where $T$ denotes one (or a few) gradient steps on the DPO objective using pairs drawn according to $\mathbf{w}$. For a small learning rate, the update direction is well approximated by a convex combination of objective-wise descent directions:

$$d(\theta; \mathbf{w}) \approx -\sum_{i=1}^n w_i \, \nabla a_i(\theta), \qquad \theta^+ \approx \theta + \eta \, d(\theta; \mathbf{w}).$$

Fix the current parameter $\theta$ and consider any weight vector $\mathbf{w} \in \Delta^{n-1}$. For each coordinate $i$,

$$a_i(\theta + \eta \, d(\theta; \mathbf{w})) \approx a_i(\theta) - \eta \sum_{j=1}^n w_j \, \nabla a_i(\theta)^\top \nabla a_j(\theta).$$

Writing $g_i(\theta) = \nabla a_i(\theta)$ and $G(\theta) = [g_1(\theta), \ldots, g_n(\theta)] \in \mathbb{R}^{p \times n}$, we have

$$\nabla a_i(\theta)^\top d(\theta; \mathbf{w}) = -\, g_i(\theta)^\top G(\theta) \, \mathbf{w}.$$

If there exists $\mathbf{w}$ such that $g_i(\theta)^\top G(\theta) \, \mathbf{w} > 0$ for all $i$, then moving along $d(\theta; \mathbf{w})$ simultaneously reduces all coordinates of $\mathbf{a}$ to first order, which means a better sampling weight still exists. Conversely, if for some tolerance $\varepsilon > 0$ one has for all $\mathbf{w} \in \Delta^{n-1}$

$$\max_i \left\{ -\nabla a_i(\theta)^\top d(\theta; \mathbf{w}) \right\} \leq \varepsilon,$$

then no convex combination of objective gradients yields a uniformly improving direction beyond $\varepsilon$. This condition is the first-order certificate that no strictly better sampling weight exists at $\theta$.

The above criterion is equivalent to Pareto stationarity. Since $d(\theta; \mathbf{w}) = -G(\theta)\mathbf{w}$ and

$$\left\| G(\theta)\mathbf{w} \right\|^2 = \sum_{i,j} w_i w_j \, g_i(\theta)^\top g_j(\theta),$$

the existence of $\mathbf{w}^\star$ with $\|G(\theta)\mathbf{w}^\star\| \leq \kappa\varepsilon$ implies that the zero vector lies, up to $\kappa\varepsilon$, in the convex hull of $\{g_i(\theta)\}_{i=1}^n$, which is the KKT-type first-order condition for an $\varepsilon$-Pareto stationary point. If

the zero vector is not in a neighborhood of that convex hull, choosing the maximizing $\mathbf{w}$ yields a common-descent direction that contradicts the no-better-weight condition.

PRISM's outer loop is

$$\theta^{(t+1)} = T(\theta^{(t)}, W(\mathbf{a}(\theta^{(t)}))), \qquad \mathbf{a}^{(t+1)} = \mathbf{a}\left(\theta^{(t+1)}\right).$$

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

. Among multi-objective DPO-style methods, PRISM also attains the lowest average time (about 0.86 versus roughly 0.89 for MODPO and 0.94 for MO-ODPO) with shorter plans on average, without sacrificing correctness (cross-task accuracy about 0.848 versus 0.825 and 0.814). In short, PRISM stays on a favorable region of the Pareto surface: high accuracy, near-zero wrong actions, and reduced latency and steps.

Overall, these results show that once distractor actions are introduced, preference learning over objective-gap directions is critical. By combining golden diagnostics with adaptive weighting and cosine-oriented sampling, PRISM consistently stays on a favorable region of the Pareto surface—high accuracy, few or no wrong actions, and lower time and step costs—outperforming alternative multi-objective DPO approaches on the combined efficiency metrics while maintaining strong correctness.

## 4.4 WEIGHT EVOLUTION AND EXPLORATION UNDER THE PARETO FRONTIER

As shown in Fig. 2, we plot the evolution of PRISM's sampling weights while probing the Pareto front on each dataset. Each panel is a ternary projection of the four-objective simplex: the three displayed axes correspond to the weights on accuracy, time, and step, and the weight on wrong is implicitly determined as one minus the sum of the other three. Polylines connect checkpoints across three epochs, revealing how attention over objectives shifts as golden diagnostics update. Because weights adapt toward under-optimized directions, the sampler focuses on comparisons most aligned

Table 2: Comparison between PRISM and other baselines in multi-objective optimization processing

| Method | Object | Question Answering | | Coding | | Math Reasoning | | Average | GPU-H |
|---|---|---|---|---|---|---|---|---|---|
| | | HotpotQA | DROP | HumanEval | MBPP | GSM8K | MATH | | |
| Scoreflow | Acc% | 0.7842 | 0.7854 | 0.9137 | 0.7388 | 0.8784 | 0.5404 | 0.7735 | 1.4667 |
| | Time(s) | 0.7413 | 0.7549 | 2.0504 | 0.7837 | 0.7027 | 0.8021 | 0.9725 | |
| | #Step | 3.3147 | 2.5641 | 6.4512 | 4.2420 | 10.2000 | 6.5214 | 5.5489 | |
| | #Wrong | 1.7150 | 1.5980 | 1.4329 | 1.3860 | 1.5360 | 0.9080 | 1.4293 | |
| MORLHF | Acc% | 0.7215 | 0.7461 | 0.8223 | 0.6947 | 0.8169 | 0.4810 | 0.7138 | **1.3534** |
| | Time(s) | 0.8006 | 0.8002 | 2.2554 | 0.8386 | 0.7378 | 0.8743 | 1.0512 | |
| | #Step | 3.5467 | 2.6923 | 7.0963 | 4.5814 | 11.1180 | 6.9112 | 5.9910 | |
| | #Wrong | 1.8865 | 1.7099 | 1.5619 | 1.4553 | 1.6589 | 0.9625 | 1.5392 | |
| MO-ODPO | Acc% | 0.8194 | 0.7933 | 0.9320 | 0.8311 | 0.9004 | 0.6083 | 0.8141 | 1.8911 |
| | Time(s) | 0.7353 | 0.7105 | 2.2493 | 0.7378 | **0.5510** | 0.6291 | 0.9355 | |
| | #Step | 2.8352 | 2.2560 | 3.7622 | 2.9840 | 6.3540 | 4.7840 | 3.8292 | |
| | #Wrong | 0.1872 | 0.1240 | 0.3659 | 0.1140 | 0.8640 | 0.4800 | 0.3559 | |
| MODPO | Acc% | 0.8239 | 0.8453 | **0.9571** | 0.8132 | 0.8964 | 0.6133 | 0.8249 | 23.5716 |
| | Time(s) | 0.7178 | 0.7249 | 1.8732 | 0.7630 | 0.6246 | 0.6166 | 0.8867 | |
| | #Step | 2.3738 | 1.8501 | 4.7317 | 3.0140 | 5.0368 | 4.0780 | 3.5141 | |
| | #Wrong | 0.0276 | **0.0000** | 0.0932 | 0.1074 | 0.0894 | 0.0629 | 0.0634 | |
| PRISM | Acc% | **0.8692** | **0.8773** | 0.9489 | **0.8389** | **0.9264** | **0.6269** | **0.8479** | 1.8475 |
| | Time(s) | **0.7136** | **0.6928** | 1.8432 | **0.7335** | 0.5619 | **0.5990** | 0.8573 | |
| | #Step | **1.9389** | **1.8120** | **3.1281** | **2.5880** | 3.3540 | 4.0020 | 2.8038 | |
| | #Wrong | **0.0000** | **0.0000** | **0.0006** | **0.0024** | **0.0000** | **0.0376** | **0.0068** | |

with current deficiencies, which in turn increases the scalarized score at each checkpoint and guides the policy toward regions consistent with Pareto balance.