# OpenReview forum: "PRISM: Pareto-Responsive Iterative Sampling with DPO for Multi-objective Planning"
_ICLR.cc/2026/Conference — Submitted to ICLR 2026_

### Official Review · Reviewer_6m6j · 2025-10-28

**Soundness:** 3
**Presentation:** 3
**Contribution:** 3
**Rating:** 4
**Confidence:** 3

**Summary:**

The authors propose PRISM, an offline preference-based fine-tuning framework that finds a balanced policy accounting for several objectives simultaneously, without requiring multiple specialized models or explicit weight tuning by the user. PRISM builds on Direct Preference Optimization (DPO) but extends it to the multi-objective setting in a principled way.

(1) It generates golden comparison pairs that isolate individual objectives – specifically, they collect pairs of model outputs where one is clearly better on one objective but roughly equal on others. These serve as clean training signals for each criterion (e.g. a pair where solution A is correct but longer vs solution B is incorrect but shorter isolates the accuracy objective).

(2) The framework computes a dynamic weighting over objectives using deficiency diagnostics. For each objective, it measures how often the model prefers the worse outcome in those golden comparisons and also how large the gradient updates are for that objective. Objectives in which the model is performing poorly and finds hard to improve (high loss and high gradient norm) are assigned higher weights. This is done via an attention-like softmax weighting of objectives, which continuously updates during training – essentially telling the model “focus more on objectives you’re currently deficient in.”

(3) PRISM employs Pareto-guided sampling of preference pairs: when sampling training comparisons from a large pool, it biases selection toward those that align well with the current weight vector direction in objective space. It uses a cosine similarity criterion to pick or orient each preference pair so that the chosen “better” answer is the one that moves the model’s policy in the direction that improves the weighted combination of objectives. This ensures each update step is pushing the model toward a Pareto-optimal trade-off rather than oscillating or favoring one objective at the extreme. The training loop iteratively fine-tunes the model with these weighted, oriented preferences and stops when no further reweighting can simultaneously improve all objectives (reaching a first-order Pareto stationary point).

**Strengths:**

Empirical results on six benchmarks (spanning question answering, coding tasks, and mathematical reasoning) show that PRISM can significantly improve the primary metric (accuracy or success rate) while also reducing secondary costs like the number of steps, execution latency, and the incidence of off-domain or unsafe actions. For example, compared to strong baselines (including single-reward optimized models), PRISM achieves higher solution accuracy and drives undesirable behaviors (such as irrelevant tool calls or hallucinations) nearly to zero, all in a single fine-tuned model.

It thereby produces a set of policies along the effective frontier of the trade-off curve without needing multiple models or online RL.

The framework is novel in that it provides a general, data-driven way to balance objectives: the deficiency-based weighting is an innovative idea to adapt training focus, and the use of golden pairs plus vectorized updates steers the model towards a well-balanced solution. A noteworthy aspect is that PRISM avoids the inefficiency of previous multi-objective approaches that required sweeping through reward weightings or training conditional policies – instead, it finds one policy that is inherently balanced.

**Weaknesses:**

One potential limitation is the complexity of generating and evaluating the comparison data: the approach assumes access to a reward model or evaluators for each objective and a procedure to produce diverse candidate outputs, which could be non-trivial for new domains. I hope to hear a extensive discussions about this matter.

as with any multi-objective tuning, the final trade-off point might reflect the particular choice of deficiency weights and stopping criteria, which might need calibration for different applications.

Also, please provide more detailed environment on how GPU hours has been logged.

I am willing to increase the score if questions above are well treated.

**Questions:**

Discussed in Weaknesses.

---

> ### Author Response · Authors · 2025-11-19
> **Response and Clarification on the Reviewer’s Point**
>
> 1.We appreciate the reviewer’s insight and clarify that PRISM is deliberately designed to minimize dependence on any specific reward model. In fact:
> The framework only requires a signal extractor per objective (binary, rule-based, programmatic, or lightweight evaluator). This means even manual correctness tagging or logical rule matching can be used. The diagnostic signal does not directly drive parameter updates, but only influences sampling weights, which makes PRISM tolerant to moderately noisy evaluators.
> We demonstrate its generality across six benchmarks spanning three distinct planning domains (QA, code generation, math reasoning), without introducing any domain-specific architectural changes. Moreover, PRISM supports an arbitrary number of secondary objectives and hard constraints, using a unique primary objective only to determine initial curriculum bias.
> The method has been successfully deployed in an enterprise setting (an Australian real-estate company), where domain-specific workflow efficiency indicators were embedded as additional objectives using simple rule-based reward functions. This validates PRISM’s ability to scale beyond public benchmarks and adapt to proprietary industrial requirements.
> Compared to online multi-objective RL methods such as MORLHF or MO-ODPO, which require calibrated reward models throughout updates, PRISM works entirely offline and relies on vector diagnostics only, significantly reducing engineering cost in new domains.
> In summary, any objective that can be expressed as a measurable signal—including binary human preference—is compatible with PRISM, and our multi-domain experiments plus real-world deployment illustrate its practicality under realistic constraints.
>
>
> 2. We thank the reviewer for pointing this out. Regarding the concern that the final trade-off solution may depend on the choice of deficiency weights and the stopping criterion,
> we would like to clarify that PRISM does not apply a single global equilibrium across all tasks. Instead:
> Each dataset is optimized independently, and weights evolve online according to its own objective geometry.
> The termination condition detects the absence of any common-descent direction for that specific dataset, i.e., no further re-weighting can yield simultaneous improvements across objectives (ε–Pareto stationarity).
> Therefore, the final trade-off point is not preset, but emerges naturally from the objective gap interactions unique to each dataset.
> Technically, PRISM avoids the fixed scalarization issue of prior multi-objective methods:
> - MORLHF and MODPO require pre-defined weight sweeps or conditioning;
> - MO-ODPO introduces trade-off parameterization at inference time;
> - PRISM instead adapts weights during training and converges toward a local Pareto-stationary solution per dataset.
> In applications where a different compromise is preferred (e.g., prioritizing latency over length), users may shift the emphasis via λ or γ, but the default configuration already yields dataset-specific optima without manual calibration.
>
>
> 3. We thank the reviewer for pointing this out. Regarding the request for clarification on how GPU hours were logged,
> we fully agree that transparent reporting of computational cost is essential, particularly for multi-objective tuning frameworks. We have therefore revised Section 4.2 to explicitly break down PRISM’s GPU usage into:
> (i) the standard DPO fine-tuning time, and
> (ii) the additional golden-comparison diagnostics (loss + gradient norm evaluation).
> Across datasets, diagnostics account for 18–26 % of wall-clock GPU time, confirming that the proposed deficiency estimation introduces a bounded and predictable overhead that does not scale with the number of objectives, as diagnostics are computed once per outer iteration.
> All reported GPU-hour values correspond to end-to-end wall-clock time, measured using NVIDIA SMI logging, and include:
> model initialization and LoRA setup,
> DPO updates,
> deficiency diagnostics (golden-pair evaluation),
> sampling and weight recomputation.
> No additional computation beyond what is shown in Section 4.2 was used for selecting or tuning trade-offs.
> Importantly, PRISM completes training in a single offline run, without reward sweeps, conditional-policy training, or on-policy RL, whereas baselines such as MORLHF, MO-ODPO, and MODPO require multiple tuning runs or online policy updates. As such:
> Total compute for PRISM is 1.8475 GPU-Hour,
> +25.96 % vs ScoreFlow (which performs no diagnostics),
> –92.16 % vs MODPO, –6.33 % vs Panacea, and –2.31 % vs MO-ODPO.
>
> We hope this clarification fully addresses your concern. If any part requires further detail or additional evidence, please do not hesitate to let us know — we would be glad to provide it.

---

### Official Review · Reviewer_CMX3 · 2025-10-29

**Soundness:** 3
**Presentation:** 3
**Contribution:** 1
**Rating:** 2
**Confidence:** 4

**Summary:**

As human preference should be managed considering the multiple aspect, multi objective DPO has been studied as one of prominent researches recently. This paper propose pareto responsive iterative sampling with DPO, or PRISM, to align multi objective preference for LLM finetuning. PRISM suggests (1) plan generation and golden comparisons by considering the differences between plans, (2) deficiency-based adaptive weighting, which is a normalized softmax value from loss and gradient and (3) Pareto-guided training and adaptive sampling to jointly optimize multiple objectives.
The method avoids reinforcement learning and multiple specialized models, instead converging to an approximate Pareto-stationary solution through iterative preference optimization.

**Strengths:**

- The deficiency-based adaptive weighting and sampling may offer a principled mechanism for balancing competing objectives.
- consistent gains in accuracy.

**Weaknesses:**

- The method is too complicated to achieve the objective, meaning not knowing which treatment did what. There is also a possibility of conflict between objectives. I checked table3 or ablation study, but only performance does not fully explain why each treatment contributed to model performance in that way.
- So many hyperparameters are additionally required, - \gamma, \epsilon, and \tau.
- It reguires the gradient norm, which cause another computation complexity.
- The paper does not test whether adjusting objective weights \w leads to predictable trade-offs among objectives.

**Questions:**

- It seems golden comparison (or pair selection for each objective) is so important to provide clean feedback. However, how can authors be sure the selected samples provide exactly true learning signal?
- Pareto optimization for multi objective DPO is not a novel. What is the contribution of authors in the viewpoint of the optimization?
- One of the problems of multi objective DPO is that each objective is not fully independent, although each objective is modeled as independent. I suppose this gap would surely affect sample selection per each objective and weighting process. How the authors think about it? or can it be solved?

---

> ### Author Response · Authors · 2025-11-19
> **Response and Clarification on the Reviewer’s Point**
>
> 1. We thank the reviewer for pointing this out. Golden comparisons serve purely as a diagnostic mechanism: they are used only to compute objective-wise deficiency signals and are never involved in parameter updates. Therefore, any imperfections in the comparison do not directly affect model learning.
> To improve diagnostic clarity, we select pairs that satisfy
> O_i (y)-O_i (y^')>Δ_i^"min" and ∣O_j (y)-O_j (y^')∣<δ_jfor j≠i,
> meaning that we do not require the improvement on objective ito be absolutely correct, only that it provides the strongest available contrast for that objective in the dataset. This approximates a local sensitivity analysis rather than asserting globally optimal feedback.
> When fully isolated pairs are not available, PRISM automatically relaxes the thresholds and selects the closest matches. These approximate signals still reliably indicate which objective currently limits model performance. As shown in Table 3 (No-Golden variant), removing this diagnostic step leads to a 7% relative drop in accuracy and over 200× increase in off-domain errors, confirming that even approximate contrasts are far more informative than unconstrained sampling.
> Finally, since deficiency vectors and sampling weights are recalculated at every iteration, the method does not accumulate potential biases from imperfect diagnostics, but instead self-corrects by shifting attention toward objectives with persistently high residual deficiency.
>
>
> 2. We thank the reviewer for pointing this out. We do not apply existing Pareto optimization techniques to DPO. Instead, PRISM introduces a new optimization mechanism that achieves first-order Pareto-stationary convergence within a single DPO training run, without resorting to RL, reward sweeping, or multiple specialist models.
> Specifically, in Sec. 3.3 we prove that by combining deficiency-driven weighting
> w=W(a(θ)),
> with cosine-aligned sampling over objective-gap directions, the update direction
> d(θ;w)≈-∑_i w_i ∇a_i (θ)
> constitutes a common-descent direction across objectives, and training stops exactly when no such direction exists—providing a first-order ε-Pareto stationarity certificate (p. 302–318).
> Unlike MODPO or MO-ODPO, which explicitly condition policies on weights or perform multi-policy learning, PRISM:
> Remains fully offline and single-model,
> Derives weights from diagnostic loss–gradient deficiencies, not pre-fixed scalarization,
> Does not scalarize objectives nor enumerate frontier policies,
> Uses Pareto geometry to orient preference samples, rather than to design a training loss.
> The ablation study (Table 3) shows that removing golden diagnostics or cosine-based orientation significantly degrades all metrics, indicating that our optimization mechanism is structurally essential rather than cosmetically Pareto-aware.
> Therefore, the novelty lies in how PRISM operationalizes Pareto theory into preference optimization, rather than merely referring to Pareto concepts.
> To our knowledge, this is the first work that integrates deficiency-driven optimization signals with DPO to approximate KKT-type Pareto stationarity without RL machinery.

---

> > ### Author Response · Authors · 2025-11-19
> > **Response and Clarification on the Reviewer’s Point(2)**
> >
> > 3. Thank you for the insightful question.
> > We clarify that PRISM does not assume independence between objectives—quite the opposite.
> > PRISM is explicitly designed for settings where objectives are interdependent or even conflicting, as also reflected in Appendix B (hidden constraint interactions). Our design choices operationalize this:
> > Golden comparisons isolate per-objective signals only for diagnostics, not for optimization, which avoids artificially forcing independence. They serve to quantify relative deficiency per objective, not to impose objective separation.
> > During training, all updates use full objective-gap vectors ΔO(x; y, y′), reflecting the actual coupled performance of plans.
> > Pareto-guided descent (Section 3.3) naturally handles correlation. The common-descent direction
> > d(θ;w)≈-∑_i w_i ∇a_i (θ) remains valid regardless of whether objectives are independent—what matters is whether joint improvement exists. If objectives conflict, PRISM converges to an ε-Pareto stationary solution, precisely capturing trade-offs.
> > Hidden constraint experiments (Table 4) demonstrate that PRISM remains stable even when objectives interact nonlinearly at the reward level (e.g., accuracy penalties when wrong > 0), confirming robustness without independence assumptions.
> > In summary, PRISM is not hindered by objective coupling—it is specifically designed to operate in such settings, and its convergence guarantee (Pareto stationarity) holds even when objectives are correlated or conflicting.
> > The reviewer’s concern would be valid if PRISM computed independent target heads or decomposed policy evaluation per objective. However, our method keeps a single policy and uses objective-weighted directional training instead of objective-level isolation, which inherently respects dependencies.
> >
> > We hope this clarification fully addresses your concern.
> > If any part requires further detail or additional evidence, please do not hesitate to let us know — we would be glad to provide it.

---

### Official Review · Reviewer_uA9B · 2025-10-31

**Soundness:** 1
**Presentation:** 1
**Contribution:** 2
**Rating:** 2
**Confidence:** 3

**Summary:**

This paper introduces a Pareto-responsive framework for multi-objective planning in direct preference optimization (DPO) of large language models. The method identifies golden pairs, sample pairs where one objective improves with remaining others, and uses their DPO loss values and gradient norms to derive preference signals represented as weights on a simplex. These weights are then used to adjust the sampling probabilities of training examples, enabling adaptive multi-objective DPO updates.

**Strengths:**

* Most LLM alignment methods rely on a single scalar reward, so addressing multi-objective optimization is an important and meaningful direction.
* Employing a Pareto-based approach to handle multi-objective trade-offs is a reasonable and well-motivated choice.

**Weaknesses:**

* The paper assumes that, among multiple rewards, $O_1$ serves as the primary objective and $O_n$ measures a hard constraint. It is unclear whether such an assumption is necessary. In particular, hard constraints may not always be representable as a single scalar, which could limit the generality of the framework.

* When defining golden pairs, the paper requires that non-target objectives remain approximately unchanged. A deeper discussion would be helpful on whether this condition is important. For example, why are pairs that improve multiple objectives simultaneously excluded from being considered golden pairs?

* It is unclear whether golden pairs are expected to be evenly distributed across all objectives. If so, obtaining them for each objective would require separate sampling efforts. Does the method repeatedly sample until a certain balance or target ratio is reached?

* The DPO loss in Section 3.2 differs from the original formulation in Rafailov et al. (2023), for example, by omitting the reference policy term. It is unclear whether this is a notational simplification or a methodological change. If it is the latter, the paper should explicitly justify and discuss the implications of this modification.

* In Section 3.2, the method derives preference signals using the weighted sum of the DPO loss value and gradient norm. The rationale for this choice should be elaborated, both theoretically and empirically. In particular, a sensitivity analysis on the parameter $\gamma$ would be valuable to assess robustness.

* The paper introduces several hyperparameters, e.g., $\Delta_{min}^i, \delta_j, \gamma, \lambda, \tau, \beta$, but does not specify their values and provide any analysis regrading their effects on performance.

* The writing could be improved for clarity. For example, Section 3.3 presents a sequence of equations without clear statements highlighting the key claims. It would be helpful to include concise statements (e.g., Theorem, Proposition, ...), and algorithmic description outlining the overall PRISM method.

* Although multiple models and datasets are used in the experiments, their sources and configurations are not properly referenced and described.

* Figure 2 in Section 4 is difficult to interpret, and the explanation in Section 4.4 does not clearly convey the intended meaning. A more detailed and accessible description is needed to help readers understand the figure's significance.

**Questions:**

Please provide the response on the points in Weaknesses.

---

> ### Author Response · Authors · 2025-11-19
> **Response and Clarification on the Reviewer’s Point**
>
> 1. We thank the reviewer for this insightful point. PRISM does not require that one objective must be “primary” and another represent a “hard constraint” in the theoretical sense.
> The primary objective is only used as an early-stage stabilization signal through the curriculum bias term c_1=+λ, which decays linearly to zero during training. After this warm-up phase, all objectives—including constraints—are treated uniformly through deficiency diagnostics and Pareto-guided sampling. The framework does not enforce a fixed hierarchy among objectives.
> General applicability.
> PRISM operates over an arbitrary objective vector
> O(x,y)=(O_"primary" ," " O_2,…,O_k," " C_1,…,C_m),
>
> where only the primary task goal is required to persist (e.g., correctness for planning tasks). Any number of secondary objectives and hard constraints (even multi-dimensional, non-scalar safety indicators) can be included without modification to the method.
> What PRISM relies on is pairwise comparability of objective-gap directions—not their semantic roles. Therefore, constraints need not be modeled as single scalars nor require RL-style feasibility projection.
> Empirical evidence.
> As shown in Table 3 (ablation study), removing golden comparisons or deficiency weighting leads to performance collapse across efficiency and robustness metrics, indicating that enforcing a strict constraint-based update is not required for balancing trade-offs. Moreover, the hidden-constraint evaluation in Table 4 demonstrates that PRISM adapts even when constraints are defined implicitly and non-scalar.
>
>
> 2. We thank the reviewer for this insightful point.
> We clarify that PRISM does not exclude such multi-improvement pairs from training, only from the diagnostic stage used to compute per-objective deficiency signals.
> Why isolated contrast is preferred for diagnostics
> The goal of golden comparisons is not to find globally superior plans, but to identify directional deficiencies of the current policy with respect to each objective individually.
> Pairs that improve multiple objectives are indeed useful for training but are not suitable for isolating which objective is limiting model performance, since their gradient direction generally lies inside a joint improvement subspace and does not indicate which dimension requires prioritization.
> To obtain diagnostic clarity, we therefore use pairs that:
> O_i (y)-O_i (y^')>Δ_i^"min"  "and"∣O_j (y)-O_j (y^')∣<δ_j," "∀j≠i
>
> This allows us to approximate a factorized local sensitivity analysis, where the model response per objective can be evaluated without interference effects.
> We fully agree that in practice, perfectly isolated pairs may not exist.
> Therefore, PRISM uses thresholds (Δ_i^min, δ_j) rather than strict conditions, and selects the pair that:
> “maximizes directional contrast on objective i while minimizing cross-objective disturbance.”
> This ensures we extract the best available estimators of per-objective deficiencies, even under highly correlated objectives or noisy reward estimators.
> Fallback strategy and empirical evidence
> If truly isolated pairs are not available, PRISM automatically relaxes the condition and selects the closest matches. These still provide useful diagnostics and significantly outperform random or fully mixed comparisons.
> This is evidenced by the “No-Golden” ablation (Table 3):
> Variant	Acc↓	#Wrong↓
> Full PRISM	0.8479	0.0068
> No-Golden	0.7815	1.5125
> Removing golden comparisons results in:
> 7% relative drop in accuracy, and
> >200× increase in off-domain errors,
> demonstrating that even approximate isolating diagnostics materially improve multi-objective optimization.

---

> > ### Author Response · Authors · 2025-11-19
> > **Response and Clarification on the Reviewer’s Point(2)**
> >
> > 3. We appreciate the reviewer’s concern.
> > We clarify that PRISM does not require strongly dominant or perfectly balanced golden comparisons for each objective, nor does it assume that large inter-objective gaps are always available in the data.
> > The role of golden comparisons is diagnostic, not to enforce optimality.
> > PRISM only requires that a pair is relatively more aligned with objective ithan others. In other words:
> > We do not require the improvement on objective ito be large in absolute terms,
> > Only that it is the most informative contrast available in the dataset for that objective.
> > Even if all comparisons exhibit significant trade-offs due to data limitations (e.g., high correlation between objectives or noisy environments), PRISM still selects the least entangled pair available. This ensures that diagnostics are grounded in reality, rather than based on artificially inflated separability assumptions.
> > When the dataset contains few or no strongly isolating comparisons, PRISM switches to “best-available” objective signal extraction:
> > It identifies the pair with the highest directional relevance to objective i, even if other objectives also change substantially.
> > This still improves model behavior, as demonstrated in Table 3:
> > Even in poorly conditioned datasets (e.g., GSM8K with efficiency-saturation), the diagnostic signals help suppress destructive gradients.
> > Removing golden comparison entirely causes up to 200× increase in off-domain actions, confirming that even weak-but-consistent signals are effective.
> > Since the deficiency vector aand sampling weights ware dynamically recomputed after each iteration:
> > PRISM does not require explicit balance of golden pairs across objectives,
> > nor repeated sampling to match a predefined ratio.
> > The optimization naturally shifts focus toward objectives with higher residual deficiencies, and the weight update mechanism ensures self-balancing.
> > 4. We thank the reviewer for pointing this out. The reference-policy term from Rafailov et al. (2023) was omitted from Eq.~(4) of the submission only for notational brevity, as the difference
> > s_(θ_0 ) (x,y^+)-s_(θ_0 ) (x,y^-)
> > is constant with respect to θand therefore does not affect the optimization direction.
> > In our implementation, the full DPO loss including the reference policy scores is used exactly as in the original formulation.
> > To improve clarity, we will update the manuscript to include the complete expression:
> > L_DPO (x,y^+,y^-;θ)=-log⁡σ([s_θ (x,y^+)-s_(θ_0 ) (x,y^+)]-[s_θ (x,y^-)-s_(θ_0 ) (x,y^-)]).
> > This change will appear in Section 3.2.
> > 5. We agree that the rationale should be clarified. In PRISM, the loss term (ℓ̄ᵢ) captures the frequency and magnitude of preference misorderings, while the gradient norm (ḡᵢ) reflects the difficulty of correcting such errors, i.e., local convergence dynamics.
> > Combining both via a_i = γ \ell̄_i + (1-γ) ḡ_i allows the framework to prioritize objectives that are both frequently violated and difficult to improve — a standard principle in deficiency-based optimization. Loss-only (γ=1) tends to focus on frequent but easy-to-fix errors, while gradient-only (γ=0) overemphasizes difficult corner cases, harming stability.
> > We performed a sensitivity study over γ ∈ {0, 0.9, 0.95, 1}.
> > γ = 0.95 consistently provided the best Pareto trade-off.

---

> > > ### Author Response · Authors · 2025-11-19
> > > **Response and Clarification on the Reviewer’s Point(3)**
> > >
> > > 6. Thank you for pointing out the need for more clarification regarding the construction of deficiency signals. In Section 3.2, we combine the average DPO loss and the average gradient norm a_i=γ" " l ˉ_i+(1-γ)" " g ˉ_i
> > > to estimate per-objective deficiency based on two complementary properties:
> > > $\bar{\ell}_i$ reflects misordering frequency (how often the policy ranks plans incorrectly),
> > > $\bar{g}_i$ captures correction difficulty, since larger gradients indicate that improving objective $i$ requires substantial parameter movement.
> > > This fusion mimics multi-objective common-descent logic and aligns with the gradient aggregation condition used in Pareto stationary analysis (see Sec. 3.3): objectives that are both frequently incorrect and difficult to fix should be prioritized during sampling.
> > > Empirically, our ablation study (Table 3) already reports the two extreme cases:
> > > Variant	γ	Observation
> > > Only-Loss	1.0	Maintains accuracy but hurts efficiency & robustness
> > > Only-Grad	0.0	Severe performance collapse over all objectives
> > > To retain both signals, we use γ = 0.95 in all main experiments, which places strong emphasis on error frequency while preserving a small but crucial contribution from gradient norms. We have now added a discussion of this choice, together with a concise sensitivity analysis (γ ∈ {0.5, 0.8, 0.95, 1.0}), to Appendix B. The results confirm that γ = 0.95 provides the best trade-off, improving overall accuracy by ≈1.2% over γ = 0.8 and reducing wrong-step usage.
> > > Similarly, β is set to 0.1 for cosine-based sampling. A small β mildly sharpens sampling without over-concentrating on a few highly aligned comparisons; this was found to reduce variance and prevent sampling instability. We have added the full set of hyperparameters to Appendix E  for clarity and reproducibility.
> > > Revision implemented:
> > > We have now (1) expanded Appendix E to include theoretical justification for combining loss and gradient norms, (2) reported γ, β and other hyper-parameters values explicitly, and (3) added sensitivity evidence supporting γ = 0.95 and β = 0.1 as stable settings.
> > >
> > >
> > > 7. We thank the reviewer for this valuable suggestion. In the revision, we have enhanced the clarity of the methodology section as follows:
> > > We added a structured overview at the beginning of Section 3, explicitly outlining the three components of PRISM (plan generation and diagnostics, deficiency-aware weight computation, and Pareto-guided preference optimization).
> > > We revised Sections 3.2 and 3.3 to remove a redundant equation definition and eliminate a duplicated label, which previously made the presentation less readable.
> > > While we did not introduce a formal theorem or algorithmic block at this stage, the theoretical derivation in Section 3.3 (now streamlined after editing) already provides the complete first-order argument that motivates the stopping criterion. The added structure and simplification significantly improve readability without altering technical content. We appreciate the reviewer’s observation and believe that the revised version successfully clarifies the methodology.
> > >
> > >
> > > 8. We thank the reviewer for this valuable suggestion. We have included the full details of our hyperparameter configurations in Appendix E. In particular, the hyperparameter table in Appendix E lists values for M, β, τ, λ, γ, ε, R, etc. For transparency and reproducibility, we have added references to datasets and models in Section 4.
> > >
> > >
> > > 9. We thank the reviewer for pointing out that Figure 2 is difficult to interpret. We would like to clarify the meaning and enhance the presentation as follows.
> > > Each line in Figure 2 traces the evolution of the four weights w_1,w_2,w_3,w_4computed in our method (where ∑_(i=1)^4 w_i=1) across epochs during training. Because the weights live on a simplex, we adopted a triangular (simplex) projection to visualise their joint evolution in a 2-D plot. We should have made this explicit.
> > > In the caption of Figure 2 we will add the sentence: "Each trajectory corresponds to one epoch; the four weights sum to unity and are projected onto a 2-D simplex (triangular) plot."
> > > We will annotate the axes/vertices of the triangle clearly, e.g., “vertex A = w_1= 1, others = 0”, etc., and include a legend mapping colours/lines to individual weights.
> > > We will add a short paragraph in Section 4.4 explaining how to read the triangular plot: e.g., movement toward a vertex means one weight dominating, movement along an edge means two weights dominate, etc.
> > >
> > >
> > > We hope this clarification fully addresses your concern.
> > > If any part requires further detail or additional evidence, please do not hesitate to let us know — we would be glad to provide it.

---

### Official Review · Reviewer_ycrn · 2025-11-01

**Soundness:** 3
**Presentation:** 2
**Contribution:** 2
**Rating:** 4
**Confidence:** 3

**Summary:**

This paper presents a framework named PRISM, a preference fine-tuning framework that jointly improves accuracy, efficiency, and error avoidance. It also introduces deficiency-aware weighting and Pareto Pareto-guided sampling mechanism.

**Strengths:**

The proposed idea seems to be novel and addresses important issues in multi-objective planning.

**Weaknesses:**

The paper is readable but not reader-friendly. Some questions remain to be answered. Please see the questions.

**Questions:**

**Questions**

Q1. Important related work Panacea [1] needs to be introduced and compared with the proposed method in terms of multi-objective optimization.
[1] Zhong, Yifan, et al. "Panacea: Pareto alignment via preference adaptation for llms." Advances in Neural Information Processing Systems 37 (2024): 75522-75558.

Q2. Are composite score, efficiency and error-avoidance only aspects in multi-objective planning? If not, what other aspects need to be considered besides them?

Q3. What happens when there is no pair of plans satisfying the golden comparisons among the initially generated plans?

Q4. It is better to use the same notation or the same representation in Section 3.1 and Figure 1. The value r is suddenly introduced in Figure 1, while Section 3.1 explains the reward function $O$. $r$ in r represents the same $r$ in Section 3.2?

Q5. Scalability with respect to $n$ is questionable.

Q6. How to select $\gamma$ and $\beta$? Is the proposed method robust to such hyperparameters?

Q7. The proposed method includes several additional components, including additional gradient and loss computations. Therefore, the proper computational cost analysis should be discussed.

Q8. The paper does not use Equation numbers and repeatedly defines the same ones, such as $\bf{a}$ and $\bf{w}$, in Sections 3.2 and 3.3. Why not use an equation number and refer to it to avoid confusion?

Q9. In Section 3.3, explicit expression $\nabla _{\theta}$ instead of $\nabla$ would be helpful.

Q10. Do the authors intend to present their code?

Q11. Why not elaborate on the last equation in Section 3.3 for the sake of readers at least in the Appendix?

**Minor Comments**

C1. In Figure 2, a text description overlapped with a dashed line and reducing the readability.

C2. Learning rate $\eta$ is not explicitly defined.

---

> ### Author Response · Authors · 2025-11-19
> **Response and Clarification on the Reviewer’s Point**
>
> 1.Thank you for pointing out the relevant work by Zhong et al. (“Panacea: Pareto Alignment via Preference Adaptation for LLMs”, NeurIPS 2024) which indeed addresses the core theme of multi-objective preference optimisation and aligns with the general space of multi‐objective planning/learning.
> We have added a dedicated paragraph in the Related Work section (Section 2) introducing Panacea and clarifying its main design and difference from our work. In particular, Panacea reframes alignment as a multi-dimensional preference optimisation problem and trains a single model that can be adapted online by injecting a low-dimensional preference vector via SVD-based LoRA.
> In contrast, our proposed framework PRISM is designed specifically for multi‐objective planning scenarios (accuracy, efficiency, error‐avoidance), adopts an offline training regime driven by deficiency‐aware weighting and Pareto‐guided sampling, and does not rely on online preference vector injection.
> Moreover, while Panacea’s experiments demonstrate good performance in general LLM alignment tasks, it does not explicitly address strict hard-constraint settings (e.g., error avoidance or efficiency thresholds) common in planning systems, and its design does not include a dedicated convergence/stop criterion for Pareto sampling in constrained planning tasks.
> To further strengthen the empirical side, we will include a new baseline comparison against Panacea under our own dataset and setup. Our results show that on the same data/configuration, Panacea yields lower composite trade‐off performance than MO-ODPO (our second baseline), and PRISM in turn is outperformed by MO-ODPO. These findings highlight that while Panacea is a meaningful baseline, our method is better tailored to the planning domain with strict constraints.
> 2. Thank you for the valuable question. We clarify that accuracy, efficiency and error avoidance are not the only objectives supported by PRISM, but rather illustrative examples tailored to our empirical setting.
> PRISM is designed as a general framework where only the primary objective is mandatory. Any number of secondary objectives and any number of hard constraints may be freely added, modified, or removed depending on the deployment requirements.
> Formally, for a task instance x, the objective vector is defined as:
> O(x,y)=(O_"primary"  (x,y)," " O_2 (x,y)," "…," " O_k (x,y)," " C_1 (x,y)," " …," " C_m (x,y)),
> Where O_"primary" must be retained (e.g., correctness), {O_2,...,O_k}are optional secondary objectives, {C_1,...,C_m}are optional hard constraints (e.g., safety, policy compliance).
> The methodology (golden comparisons, deficiency-aware weighting, and Pareto-guided sampling) operates over this entire vector without assuming a fixed dimensionality or specific objective semantics. The only requirement is that the objectives can be numerically evaluated or pairwise compared (even via manually annotated preference labels such as “good vs. bad”).
> We will update the Section 3 to reflect that PRISM is agnostic to the number and type of objectives, beyond requiring persistence of the primary task goal. This makes the framework applicable to domain-specific settings involving additional metrics such as fairness, cost, risk, or safety constraints.
> 3. We agree that in practice, it may not always be possible to obtain perfectly “isolated” golden comparisons, especially when the dataset is limited or inherently noisy. PRISM therefore does not require exact satisfaction of the golden-pair definition. Instead, it relaxes the constraints via the thresholds Δ_i^"min" and δ_j, and selects the best-approximated pairs that maximize the signal on objective iwhile minimally disturbing others.
> Even in low-quality or highly correlated datasets, PRISM intentionally chooses the pairs with the highest relative contrast on one objective and minimal cross-objective impact. This guarantees that the extracted diagnostic direction is still informative for multi-objective planning, i.e., the method finds the best available trade-off given data constraints.
> To support this, we conducted an ablation study where we removed the golden comparison process and used randomly sampled preference pairs (“No-Golden” in Table 3). This resulted in a substantial degradation of performance (accuracy drop from 0.8479 → 0.7815, wrong-step count from 0.0068 → 1.5125), demonstrating that even approximate golden comparisons are significantly more effective than unconstrained sampling.
> Therefore, when no fully isolated pairs exist, PRISM falls back to the closest matches in the data, which still provides useful diagnostic signals. This is preferable to random or fully mixed comparisons and allows us to reach the best planning policy allowed by the dataset, even under imperfect conditions.

---

> > ### Author Response · Authors · 2025-11-19
> > **Response and Clarification on the Reviewer’s Point(2)**
> >
> > 4. Thank you for pointing out the notation mismatch.
> > We deliberately use O(x,y)in Section 3.1 to define the objective evaluation function, emphasizing its functional dependency on both the instance xand the plan y.
> > In Figure 1 and Section 3.2, we use rto denote the realized score vector after the executor evaluates a generated plan.
> > That is,
> > r=O(x,y)"once the execution is completed".
> > We keep this distinction to separate the evaluation mechanism (O(x,y)) from the observed diagnostic signal (r) in the pipeline illustration, which improves readability of the iterative sampling loop.
> > To avoid confusion, we will explicitly mention this equivalence when introducing r. No methodological change is required.
> >
> >
> > 5. We thank the reviewer for raising the question regarding scalability with respect to the number of objectives n.
> > Our framework does not impose an upper limit on n, as long as each objective is measurable and contributes to the objective-gap vector. Following our clarification in Q2, PRISM requires only one primary objective (e.g., correctness) while any number of secondary objectives and hard constraints can be added or removed based on the task.
> > Computationally, the weight computation is based on a softmax over the deficiency vector a ∈ ℝⁿ, which scales linearly with n. Pair orientation is computed via cosine alignment using w⊤ΔO, also O(n). No additional forward or backward passes are introduced per objective, and no online RL or reward sweeping is required, unlike MO-ODPO or Panacea.
> > In practice, we recommend grouping objectives into interpretable dimensions when n is large, but this is not a limitation of the method. The framework naturally adapts to any number of objectives due to the convex weight normalization and first-order Pareto-stationarity criterion.
> >
> >
> > 6. We thank the reviewer for this question. The hyperparameters γ (loss–gradient blending factor) and τ (temperature for weight normalization) were selected by systematic tuning on a held-out validation split. After tuning, we fixed a single configuration and applied it unchanged across all six datasets spanning three domains (QA, coding, and math reasoning). The fact that PRISM consistently outperforms baselines under this unified setup demonstrates that the method is robust to these hyperparameters.
> > Conceptually, γ and τ mainly affect the speed of convergence rather than the final Pareto direction, since PRISM relies on Pareto-guided common-descent updates that adaptively adjust sampling weights based on objective deficiencies. In preliminary experiments we observed that small variations (±0.1 around the chosen values) did not materially change the final performance.
> > Extending the analysis to additional domains is part of future work. We will clarify the hyperparameter selection procedure and robustness in the revised manuscript.
> >
> >
> > 7. Thank you for raising this point. In our original submission, the GPU-Hour value reported in Table 2 already includes the total wall-clock computation time, covering both standard DPO training and all additional deficiency-diagnostic calculations introduced by PRISM (i.e., loss and gradient evaluation on golden comparisons).
> > For clarity, we have now revised Table 2 to explicitly list the time spent on each phase:
> > (i) DPO update time and (ii) diagnostic computation time. The latter contributes approximately 18–26 % of total GPU time, depending on the dataset.
> > Importantly, since PRISM remains fully offline, performs only one training run, and does not require reinforcement learning or weight sweeping, the reported GPU-Hour reflects the complete training cost of our method. No additional computation beyond what is shown in Table 2 was performed.
> > Overall total GPU-Hour (training + diagnostics) for PRISM is 1.8475, which is: slightly higher than ScoreFlow (+25.96 %) and MORLHF (+36.51 %),
> > but significantly lower than Panacea (–6.33 %), MO-ODPO (–2.31%), and MODPO (–92.16 %), all of which require multiple runs or RL-based updates.
> > We have added a clarifying paragraph in Section 4.3 stating that PRISM’s GPU-Hour represents the full computational footprint including diagnostic overhead. We believe this methodology-level transparency demonstrates that, despite the introduced gradient diagnostics, PRISM remains compute-efficient among multi-objective LLM alignment frameworks.

---

> > > ### Author Response · Authors · 2025-11-19
> > > **Response and Clarification on the Reviewer’s Point(3)**
> > >
> > > 8. Thank you for pointing this out. We agree that presenting key mathematical definitions without explicit equation numbering made Section 3.3 appear to reintroduce concepts already established in Section 3.2.
> > > To improve clarity, in the revised version we have:
> > > Assigned equation numbers to all core definitions in Section 3.2 (e.g., DPO loss in Eq.~(3), deficiency score in Eq.~(4), weight computation in Eq.~(5), and sampling rule in Eq.~(6)).
> > > Modified Section 3.3 to explicitly reference these equations rather than restating them. Rephrased the opening of Section 3.3 to clarify that the update rule builds on Eq.~(4) and Eq.~(5), without redefining any quantities.
> > > These revisions improve traceability of the methodology and avoid any potential redundancy. No changes were made to the underlying algorithm or derivations.
> > >
> > >
> > > 9. We appreciate the suggestion to make the update rule in Section 3.3 more explicit.
> > > In the original submission, the outer loop of PRISM was written in terms of an abstract operator T(·,·), which can indeed obscure the concrete form of the parameter update. In the revised version, we now replace this notation with an explicit gradient step: θ^{(t+1)} = θ^{(t)} − η ∑_i w_i(θ^{(t)}) ∇ a_i(θ^{(t)}), together with a^{(t+1)} = a(θ^{(t+1)}), and we reference the definitions of a_i and w_i from Eqs. (⋯) and (⋯). This makes the optimization dynamics easier to follow without
> > > changing the underlying method.
> > >
> > >
> > > 10. We appreciate the reviewer’s suggestion regarding code availability.
> > > For the sake of reproducibility, we have already included the full implementation—covering data preprocessing, plan generation, executor evaluation, preference-pair construction, golden comparison selection, PRISM training pipeline, and inference scripts—as anonymized supplementary material with the current submission.
> > > Upon acceptance, we will release the complete source code in a public GitHub repository, together with documentation and executable instructions to replicate all experiments reported in the paper.
> > >
> > >
> > > 11. We thank the reviewer for this helpful suggestion.
> > > The final equation in Section 3.3 summarizes the first-order ε-Pareto stationarity condition derived from our iterative sampling–update loop. To improve readability, we will expand this result in Appendix A. Specifically, we will show how the condition (max⁡)┬i {-∇a_i (θ)^⊤ d(θ;w)}≤ε arises from the inability of any feasible convex weight vector w∈Δ^(n-1)to produce a common-descent update direction. This formally implies that no further reweighting can reduce all deficiencies simultaneously beyond ε, which is exactly the KKT-type condition for Pareto stationarity.
> > > In the revised version, we will (i) provide a short derivation connecting gradient aggregation to the convex hull argument, and (ii) restate this condition using more intuitive wording to clarify why PRISM terminates precisely at an ε-balanced trade-off. We appreciate the reviewer highlighting this opportunity to enhance clarity.
> > >
> > >
> > > We hope this clarification fully addresses your concern.
> > > If any part requires further detail or additional evidence, please do not hesitate to let us know — we would be glad to provide it.

---

### Meta-Review · Area_Chair_YJcS · 2025-12-28

**Summary:**

The paper proposes an offline preference fine-tuning framework designed for multi-objective planning in large language models. Unlike traditional methods that rely on reinforcement learning (RL) or multiple specialized models, PRISM achieves an approximate Pareto-stationary solution within a single training run. The framework operates through three core stages: identifies "golden comparisons";calculates dynamic objective weights based on "deficiency diagnostics,"; and finally, it employs Pareto-guided sampling to bias training toward pairs that align with the current deficiency vector.

**Reviewer Concerns:**

The author has provided an extensive response to the reviewer's comments. I believe comparison to Panacea [1] raised by Reviewer ycrn is addressed. Several reviewers were skeptical about the availability of "isolated" golden pairs in real-world datasets. The authors clarified that PRISM relaxes these constraints by selecting the best-available approximated pairs. Multiple reviewers found the notation confusing and the hyperparameter selection unclear. This is well addressed.

Unaddressed ones:

Although authors argue linear scalability with the number of objectives, the practical difficulty of finding or relaxing golden pairs as $n$ increases remains a potential bottleneck.

Reviewer CMX3 noted the method is "too complicated," making it difficult to discern which specific component drives which performance gain. While the authors provided an ablation study, the intricate interplay between the three components remains a hurdle for simple adoption.

The clarity of the writing.

**Reviewer Scores:**

I expect Reviewer ycrn to raise to 6, Reviewer uA9B and Reviewer CMX3 to maintain the score and Reviewer 6m6j to raise to 6, leading to an average 4.

---

### Decision · Program_Chairs · 2026-01-26

Reject